# IR-Agent: Expert-Inspired LLM Agents for Structure Elucidation from Infrared Spectra

**Heewoong Noh**[1,✉]**, Namkyeong Lee**[1,✉]**, Gyoung S. Na**[1,2,✉]**, Kibum Kim**[1,✉]**,
Chanyoung Park**[1,✉*]

[1]KAIST, [2]KRICT

## Abstract

Spectral analysis provides crucial clues for the elucidation of unknown materials. Among various techniques, infrared spectroscopy (IR) plays an important role in laboratory settings due to its high accessibility and low cost. However, existing approaches often fail to reflect expert analytical processes and lack flexibility in incorporating diverse types of chemical knowledge, which is essential in real-world analytical scenarios. In this paper, we propose IR-Agent, a novel multi-agent framework for molecular structure elucidation from IR spectra. The framework is designed to emulate expert-driven IR analysis procedures and is inherently extensible. Each agent specializes in a specific aspect of IR interpretation, and their complementary roles enable integrated reasoning, thereby improving the overall accuracy of structure elucidation. Through extensive experiments, we demonstrate that IR-Agent not only improves baseline performance on experimental IR spectra but also shows strong adaptability to various forms of chemical information. The source code for IR-Agent is available at https://github.com/HeewoongNoh/IR-Agent.

## 1 Introduction

Spectral analysis provides critical clues for the elucidation of unknown materials (Huang et al., 2021; Li & Kang, 2020). In particular, spectroscopic techniques such as Infrared Spectroscopy (IR), Mass Spectrometry (MS), and Nuclear Magnetic Resonance Spectroscopy (NMR) are widely used for elucidating molecular structures (Li & Kang, 2020). For example, IR reveals details about chemical bonds and substructures (Griffiths, 2006), MS offers molecular weight and fragmented molecular structures (Lee, 1998), and NMR provides in-depth structural information, including stereochemistry (Li & Kang, 2020; Klein, 2013). Although IR spectroscopy lacks comprehensive chemical information such as molecular weight, stoichiometry, and stereochemical details compared to MS and NMR, it is frequently utilized in the initial phase of analysis due to its low cost, speed, and high accessibility in laboratory settings (Coates et al., 2000; Mistek & Lednev, 2018). Despite the ease and affordability of acquiring IR spectra, their interpretation remains a challenging and time-consuming task that requires extensive domain knowledge and expert experience (Varmuza et al., 1999; Jung et al., 2023).

To automate IR spectra analysis, various machine learning (ML)-based approaches have been explored. Early ML approaches for IR spectra analysis primarily focus on identifying functional groups, achieving high predictive accuracy. Specifically, convolutional neural network (CNN) architectures are employed to classify functional groups from IR spectra (Jung et al., 2023; Wang et al., 2023), and the M-order Markov property is utilized to construct IR spectrum graphs for tasks such as material class classification and functional group detection (Na & Rho, 2024). While functional group classification enables rapid and simple characterization of compounds, it remains insufficient for tasks such as material discovery or identification of unknown material, underscoring the necessity of complete molecular structure elucidation.

More specifically, molecular structure elucidation—which aims to generate the full SMILES representation of an unknown molecule—requires more comprehensive molecular information such as atomic composition, bonding information, and the connectivity of substructures, making it a substantially

---

*Corresponding author (cy.park@kaist.ac.kr)

more complex task compared to functional group classification (Xue et al., 2023). As a result, only a few recent studies have explored early approaches, including predicting molecular structures as SMILES sequences by leveraging Transformer models with chemical formula information (Alberts et al., 2024a; Wu et al., 2025), exploring reinforcement learning with IR spectra alone (Ellis et al., 2023), and extending such methods to integrate both IR and NMR spectra (Devata et al., 2024). Despite these early advances, these methods generally rely on fixed and predefined input formats, which restricts their flexibility in accommodating diverse types of chemical information.

In real-world analytical scenarios, IR spectra are often accompanied by diverse chemical information, such as atom types inferred from synthesis pathways, the number of carbon atoms, or molecular scaffolds (i.e., the molecular skeleton structure). However, existing methods struggle to flexibly incorporate such information, since accommodating new types of inputs typically requires redesigning and retraining the model (Alberts et al., 2024a; Jung et al., 2023; Devata et al., 2024). This highlights the need for a framework that can seamlessly integrate a wide range of chemical inputs.

Recently, by representing chemical information in natural language, Large Language Models (LLMs) have been effectively utilized in the field of biochemistry(Lee et al., 2024). For instance, LLMs have been used to generate molecular structures in the string representation of molecules (i.e., SMILES) from text-based descriptions (Edwards et al., 2022) and even modify molecular structures based on specified conditions (Li et al., 2024; Liu et al., 2024a). These successes highlight that an LLM-based framework is well-suited for building a more flexible and extensible IR spectrum-based structure elucidation system.

On the other hand, beyond the various types of chemical information, IR spectra analysis involves comprehensively integrating knowledge from diverse sources. Specifically, during the process, experts interpret IR absorption tables to infer local substructures and bonding patterns from peak positions (Socrates, 2004; Larkin, 2017), and retrieve structurally similar molecules from spectral databases to provide global contextual clues (Moldoveanu & Rapson, 1987). Similar to expert workflows, a successful system should accurately perform each of these tasks—extracting critical information from multiple sources—and ultimately integrate the results in a coherent reasoning process to predict the molecular structure. However, it is widely known that relying on a single LLM to perform all sub-tasks simultaneously can result in suboptimal information extraction and may be inadequate for handling complex reasoning tasks (Chen et al., 2024; Sun et al., 2023).

To this end, we propose IR-Agent, a novel LLM-based multi-agent framework specifically designed to emulate expert analytical processes and seamlessly incorporate various types of knowledge into the structure elucidation workflow based on IR spectra. Rather than relying on a single LLM to process all types of knowledge at once, our framework adopts modeling with specialized sub-agents tailored to each type of knowledge. More specifically, we design a multi-agent framework composed of the following: (1) a **Table Interpretation Expert** that performs table-guided absorption analysis to extract local structural information from the target IR spectrum; (2) a **Retriever Expert** that identifies similar spectra from spectra databases to provide global contextual structural information; and (3) a **Structure Elucidation Expert** that produces the final structure prediction by integrating the analyses from both expert agents, each contributing complementary information for complete structure elucidation. This integrative analysis within a multi-agent framework enables effective molecular structure elucidation by extracting relevant information from each knowledge source and performing collaborative reasoning. A further appeal of IR-Agent is its flexibility: when new knowledge becomes available, the system does not need a complete redesign or retraining. Instead, it can be easily extended by incorporating the additional information through updated prompts to guide the agent's reasoning process. Our main contributions in this study are as follows:

- We introduce IR-Agent, a novel multi-agent framework for molecular structure elucidation from infrared (IR) spectra. This framework models expert-driven IR spectrum analysis processes and is designed to be highly extensible.
- While each agent specializes in a specific aspect of IR spectrum analysis, their complementary roles enable an integrative analysis process, ultimately improving molecular structure elucidation from IR spectrum.
- Through extensive experimentation, we show that our proposed framework not only improves baseline performance on experimental IR spectra but also exhibits strong adaptability to diverse types of chemical information.

To the best of our knowledge, this is the first work to leverage the LLM agents framework for molecular structure elucidation from IR spectra.

## 2 RELATED WORKS

### 2.1 MACHINE LEARNING FOR IR SPECTRA ANALYSIS

Recently, ML approaches have become game changers in the field of molecular science, where traditional research has heavily relied on theory, experimentation, and computer simulation, which are often costly and time-consuming Lee et al. (2023a;b). In particular, ML approaches for IR spectra have demonstrated effectiveness in functional group identification, structural feature extraction, and molecular structure elucidation. CNNs have been applied to functional group classification (Jung et al., 2023; Wang et al., 2023), while GNNs have been used on spectrum graphs derived from the M-order Markov property for material classification and functional group detection (Na & Rho, 2024). Beyond functional groups, ML has also been extended to full molecular structure elucidation. Transformer-based models are widely used: Alberts et al. (2024a) convert downsampled IR spectra into text, and Wu et al. (2025) apply patch-based self-attention with data augmentation. Unlike these approaches, which assume access to ground-truth chemical formulas, our setting relies solely on the IR spectrum. Additionally, reinforcement learning has been applied to structure elucidation using IR spectra (Ellis et al., 2023), with subsequent work extending this approach to incorporate both IR and NMR spectra (Devata et al., 2024). More recently, large language models have been explored for structure elucidation tasks with multi-modal spectral inputs, including IR, MS, and NMR spectra (Guo et al., 2024). Unlike prior work, IR-Agent aims to incorporate expert analytical processes and adopts a multi-agent framework, which offers high architectural flexibility.

### 2.2 LLM AGENTS FOR SCIENCE

LLM agents have demonstrated strong capabilities across various scientific domains. For example, ChemCrow(Bran et al., 2023) employs an LLM agent to autonomously perform tasks typically conducted by chemists, using a range of external tools. Similarly, Coscientist(Boiko et al., 2023) autonomously handles experimental design, planning, and execution of complex experiments by integrating internet search, code execution, and laboratory automation. Moreover, LLM agents have been applied to diverse fields such as materials science(Zhang et al., 2024) and biomedical domain, including applications in drug discovery(Inoue et al., 2024) and the design of biological experiments(Roohani et al., 2024). In addition, recent work has explored the use of multi-agent frameworks to effectively tackle drug discovery tasks(Lee et al., 2025; Liu et al., 2024b), highlighting the growing interest in collaborative LLM-based systems for complex scientific workflows. While LLM agents have not yet been applied to spectra-related tasks, their integration with external tools presents high potential for extensibility and effectiveness, as demonstrated in other scientific domains. This work aims to initiate exploration in this direction by positioning LLM agents as a viable solution for spectral analysis.

## 3 PRELIMINARIES

### 3.1 PROBLEM SETUP

**Task Description.** Given an IR spectrum $\mathcal{X} \in \mathbb{R}^{1 \times L}$ of a molecule as input, where $L$ denotes the number of absorbance values corresponding to wavenumber positions, IR-Agent predicts the raw SMILES representation of the molecule, a process known as molecular structure elucidation. While some studies (Wu et al., 2025; Alberts et al., 2024a) assume that the ground-truth chemical formula is always available along with the IR spectrum, our setting considers only the IR spectrum. In practice, obtaining the exact formula of an unknown material is often unrealistic. Although mass spectrometry (MS) is commonly employed, it is costly and time-consuming (Vas & Vekey, 2004), difficult to interpret (Rolland & Prell, 2021), and still leaves the derivation of an exact formula as a highly non-trivial challenge (Böcker & Dührkop, 2016; Goldman et al., 2023). Thus, we adopt a setting where the chemical formula is not used, employing a translator that directly predicts SMILES from IR spectra. Since our framework can also incorporate supplementary *chemical information*—such as atom types, carbon counts, or scaffold structures—we further explore its applicability to more informed settings (Section 4.4). More details about analysis settings are provided in the AppendixA.2.

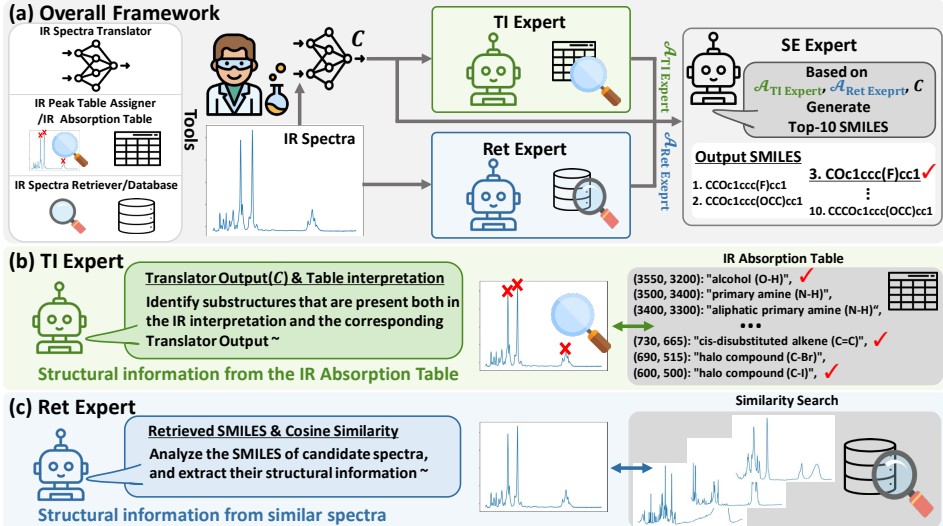

Figure 1: Overview of IR-Agent. (a) Overall framework. Given an unknown IR spectrum, IR-Agent first utilizes the IR Spectra Translator to generate candidate structures in SMILES format. The Table Interpretation (TI) Expert then extracts local structural information by referencing the IR absorption table through the IR Peak Table Assigner. In parallel, the Retriever (Ret) Expert obtains global structural features from similar spectra retrieved by the IR Spectra Retriever from a database. The Structure Elucidation (SE) Expert integrates analyses from both experts to produce the final predicted molecular structures. (b) Detailed view of the Table Interpretation (TI) Expert. (c) Detailed view of the Retriever (Ret) Expert.

**Tools.** In this paper, we specifically design tools to support task-specific analysis as follows:

- *IR Peak Table Assigner* extracts the peaks from the spectrum and finds relevant substructures from the IR absorption table.
- *IR Spectra Retriever* retrieves IR spectra from the IR Spectra Database that are similar to a given input spectrum.

**External Knowledge.** We also use external knowledge to support task-specific analysis as follows:

- *IR Absorption Table* summarizes the characteristic absorption frequencies associated with different molecular functional groups.
- The *IR Spectra Database* contains a variety of IR spectra along with their corresponding molecules in SMILES format.

Additional details on the tools and external knowledge are provided in the Appendix A.

### 3.2 IR SPECTRA TRANSLATOR

To begin with, we introduce an IR Spectra Translator, a Transformer-based model (Vaswani et al., 2017) that proposes an initial pool of SMILES candidates from a target IR spectrum. Specifically, given the target IR spectrum $\mathcal{X}$, we obtain a set of SMILES candidates $\mathcal{C}$ as follows:

$$\mathcal{C} = \{\mathbf{s}_1, \ldots, \mathbf{s_K}\} = \text{Transformer}(\mathcal{X}), \tag{1}$$

where $\mathbf{K}$ denotes the number of SMILES candidates generated by the Translator using beam search decoding. Since deriving reliable SMILES directly from thousands of real-valued IR absorbance measurements is challenging for LLMs, this module seeds the downstream reasoning process with plausible starting structures, which are subsequently expanded and revised. Additional details on the IR Spectra Translator are provided in Appendix A.2.

## 4 PROPOSED METHOD: IR-AGENT

In this section, we introduce IR-Agent, a multi-agent framework for molecular structure elucidation from IR spectra that mimics expert analytical processes through specialized expert agents. Our

framework combines off-the-shelf LLMs with analytical tools to support an integrative analysis process. It is composed of the following expert agents: (1) the **Table Interpretation (TI) Expert**, which employs an IR absorption table to identify substructures from SMILES sequences (Section 4.1); (2) the **Retriever (Ret) Expert**, which extracts structural information from retrieved spectra (Section 4.2); and (3) the **Structure Elucidation (SE) Expert**, which provides a ranked list of SMILES based on the outputs from both the Retriever Experts and Table Interpretation Experts (Section 4.3). The overall framework is presented in Figure 1.

## 4.1 TABLE INTERPRETATION (TI) EXPERT

The use of IR absorption tables is grounded in decades of experimental validation and theoretical development, offering reliable and interpretable structural insights. Importantly, this approach captures fine-grained localized structural features, such as substitution patterns, cis/trans isomerism, and conjugation, which makes it a crucial component in structure elucidation. However, effective utilization of these tables requires accurate identification of spectral peaks, which is challenging to an LLM agent because: (1) it may infer peak positions from spectral images, but only approximately and often without sufficient precision; and (2) it struggles to detect peaks directly from high-dimensional numerical absorbance data, which typically consists of thousands of values.

To address this limitation, the TI Expert agent employs the IR Peak Table Assigner tool, which extracts peaks from the spectrum by simply comparing the absorbance of neighboring wavenumbers and then assigns corresponding substructures to each peak based on its wavenumber range, by referring to the IR absorption table. An example output of the IR Peak Table Assigner is: "Peaks observed between 1200 and 1000 cm$^{-1}$ are typically associated with fluoro compounds (C–F)." Specifically, given the task-specific prompt $\mathbf{P}_{\text{TI Expert}}$, the IR Absorption Table ($\mathbf{T}$), and IR Peak Table Assigner, the agent is defined as follows:

$$\mathcal{A}_{\text{TI Expert}} = \text{TI Expert}\left(\mathbf{P}_{\text{TI Expert}}, \text{IR Peak Table Assigner}(\mathcal{X}, \mathbf{T}), \mathcal{C}\right), \tag{2}$$

where $\mathcal{C}$ denotes the SMILES candidates generated by the IR spectra Translator. $\mathcal{A}_{\text{TI Expert}}$ includes the potential substructures that can be included in the SMILES string.

Despite its utility, table-based interpretation has inherent limitations: IR spectra often contain noise, and multiple substructures may exhibit absorption within the same wavenumber region, leading to ambiguity in peak assignment. To mitigate the possible misinterpretation, we design a prompt $\mathbf{P}_{\text{TI Expert}}$ that guides the agent to compare the output of the IR Peak Table Assigner with the SMILES candidates $\mathcal{C}$, identify shared substructures, and generate a confidence level along with a brief rationale for each identified substructure (e.g., substructure $\rightarrow$ confidence $\rightarrow$ brief rationale). By doing so, the agent enhances the reliability of the table-based interpretation for target spectra. Details of IR Absorption Table, IR Peak Table Assigner, the textual prompt are provided in Appendix A.3,A.4, and E, respectively.

## 4.2 RETRIEVER (RET) EXPERT

Although the local structural information provided by the TI Expert is valuable, it is often insufficient to uniquely determine the complete molecular structure. This limitation arises because IR spectra offer vibrational information localized to specific functional groups or substructures, rather than providing a direct mapping to the full molecular structure (Coates et al., 2000; Griffiths, 2006). To overcome this limitation, we draw inspiration from the typical reasoning process of human experts, who frequently consult spectral databases to identify structurally similar reference compounds when analyzing unknown spectra (Moldoveanu & Rapson, 1987). Accordingly, we propose the Retriever (Ret) Expert agent, which leverages known molecular structures associated with similar IR spectra to provide global structural context, effectively linking local substructures to a more complete molecular structure.

Specifically, the Ret Expert agent utilizes the IR Spectra Retriever tool to identify spectra that are similar to the target IR spectrum. The IR Spectra Retriever adopts a simple yet effective approach: (1) it computes the cosine similarity between the target spectrum and all spectra in the database; (2) it then retrieves the top-$N$ most similar spectra, each associated with its corresponding SMILES structure. The output of IR Spectra Retriever for the target spectrum($\mathcal{X}$) is defined as follows:

$$\{\text{candi}_1 : \text{sim}_1, \ldots, \text{candi}_N : \text{sim}_N\} = \text{IR Spectra Retriever}(\mathcal{X}), \tag{3}$$

where $\text{candi}_i$ denotes the SMILES corresponding to the $i$-the retrieved spectrum, and $\text{sim}_i$ denotes its cosine similarity to the the target spectrum. Given task-specific prompt $\mathbf{P}_{\text{Ret Expert}}$ and the retrieval output, the agent is defined as follows:

$$\mathcal{A}_{\text{Ret Expert}} = \text{Ret Expert}\left(\mathbf{P}_{\text{Ret Expert}}, \text{IR Spectra Retriever}(\mathcal{X})\right). \tag{4}$$

The output of the Ret Expert, i.e., $\mathcal{A}_{\text{Ret Expert}}$, includes shared structural features among the top-$N$ retrieved SMILES. Given the SMILES and the cosine similarity between the target spectrum and the retrieved spectrum, the Ret Expert automatically identifies common substructures while assigning higher weight to spectra with increased similarity. These structural features provide global contextual clues that guide molecular structure reasoning. The complete prompt is provided in Appendix E.

### 4.3 STRUCTURE ELUCIDATION (SE) EXPERT

Finally, the Structure Elucidation (SE) Expert conducts integrative structure reasoning based on both $\mathcal{A}_{\text{TI Expert}}$ and $\mathcal{A}_{\text{Ret Expert}}$, as follows:

$$\mathcal{A}_{\text{SE Expert}} = \text{SE Expert}\left(\mathbf{P}_{\text{SE Expert}}, \mathcal{A}_{\text{TI Expert}}, \mathcal{A}_{\text{Ret Expert}}, \mathcal{C}\right), \tag{5}$$

where $\mathcal{A}_{\text{SE Expert}}$ includes a final ranked list of the top-$K$ predicted molecular structures. By utilizing the information provided by both agents, the SE expert agent is able to perform a comprehensive reasoning process that integrates both local and global molecular structures. Moreover, structural features consistently identified by both agents can serve as reliable cues for the SE expert agent in molecular structure elucidation.

### 4.4 INCORPORATING VARIOUS CHEMICAL INFORMATION INTO AGENT REASONING

In real-world analytical scenarios, IR spectra are often accompanied by various types of chemical information, necessitating approaches that are capable of integrating this additional information. As the IR-Agent framework is based on LLM agents, it is not constrained by fixed input formats as in conventional ML approaches, and can flexibly incorporate various chemical information in textual form. Specifically, rather than instantiating a separate agent for chemical information, we embed chemical information directly into the reasoning prompts of all the agents. Moreover, to avoid the complexity of prompt engineering, we simply append a concise sentence containing the relevant chemical information to the original prompt. This lightweight strategy reduces the cost of adding new agents and designing new prompts, while enabling each agent to perform its original task more effectively by leveraging chemical information during reasoning. Therefore, IR-Agent is applicable not only in scenarios where only IR spectral data is available, but also in cases where additional chemical information is provided, thereby enhancing the flexibility and applicability of the framework without requiring additional training or architectural modifications. We provide more details on the prompt that incorporates chemical information in the Appendix E.

### 4.5 DISCUSSION: MULTI-AGENT FRAMEWORK FOR STRUCTURE ELUCIDATION

Our IR-Agent employs a multi-agent framework in which each LLM agent is assigned a distinct sub-task within the overall structure elucidation process. Rather than employing a single LLM to manage the entire reasoning pipeline, IR-Agent distributes the workload across specialized agents with each agent focusing on a specific type of analytical reasoning, and integrates their outputs to infer the final molecular structure. Each sub-task poses unique reasoning challenges: the TI Expert performs precise local pattern recognition and chemical knowledge grounding to interpret peak–substructure mappings; the Ret Expert needs to reason over spectral similarity and extract structurally meaningful global patterns from retrieved candidates; and the SE Expert is tasked with integrating these heterogeneous insights into a coherent molecular structure. When a single-agent model attempts to perform all these sub-tasks simultaneously, it often struggles to distinguish and prioritize relevant signals for each stage. For instance, local absorption features may be misinterpreted by global context, or retrieved candidates may not be properly utilized when misleading substructure signals are present. Additionally, the increased cognitive burden of handling diverse input information (e.g., tables interpretation by peak region, retrieved SMILES) within a single context window can result in incomplete reasoning, leading to degraded predictions. To evaluate the effectiveness of our multi-agent framework in addressing these issues, we conduct a comparative analysis against its single-agent counterpart, as presented in Section 5.2.

## 5 EXPERIMENTS

### 5.1 EXPERIMENTAL SETUP

**Datasets.** In this study, we primarily use a dataset of 9,052 experimental IR spectra from the NIST database, which has been widely adopted in prior IR spectra modeling studies Jung et al. (2023); Na (2024); Wang et al. (2023). The use of experimental spectra is particularly important, as they reflect the noise, peak broadening, and variability inherent in real-world measurements, making them more representative of practical compound analysis scenarios. These spectra also reflect the types of challenges typically encountered in laboratory settings, where structural elucidation requires interpreting imperfect signals through expert knowledge and heuristics. Since our framework relies heavily on absorption table interpretation and human-like reasoning with retrieval-based search, experimental spectra provide a realistic and practical basis for evaluating its effectiveness in real-world analytical workflows. Moreover, to ensure dataset diversity, we include spectra from all phases (solid, liquid, and gas), do not exclude compounds with stereochemistry or ionic features, and impose no restrictions on heavy-atom count or the presence of mixtures. Additional dataset details are provided in Appendix B, and the performance of IR-Agent on both single compounds and mixtures is reported in Appendix C.6.

**External Knowledge.** We use the IR Absorption Table available online[1] and employ the training set as an IR Spectra Database for retrieval.

**Methods Compared.** To validate the effectiveness of IR-Agent, we compare it with the standalone **Transformer** model used as our IR Spectra Translator, showing that our framework provides additional gains in structure elucidation. We further evaluate a single-agent variant of IR-Agent, where a single LLM agent simultaneously handles all sub-tasks. To assess the impact of the underlying LLM, we vary the backbone model used in both the **single-agent** and **multi-agent** settings across **GPT-4o-mini**, **GPT-4o**, and **o3-mini**. We also consider a setting where o3-mini is used to directly generate a ranked list of 10 SMILES structures, relying solely on the input candidate set $\mathcal{C}$.

**Evaluation Protocol.** We randomly split the dataset into train/valid/test of 80/10/10%. The IR Spectra Translator is trained on this split prior to applying IR-Agent. We adopt Top-$K$ exact match accuracy as the evaluation metric to assess the effectiveness of the proposed method. This metric checks whether the correct SMILES is included among the top $K$ generated candidates, comparing structures after conversion to the InChI representation (Heller et al., 2015). We report the average performance across three independent experiments.

### 5.2 RESULTS OF STRUCTURE ELUCIDATION

**Effectiveness of IR-Agent.** As shown in Table 1, we observe the following: (**1**) Comprehensive reasoning based on the analyses from the TI Expert and the Ret Expert leads to more accurate molecular structure predictions. Given the candidate set $\mathcal{C}$ ($\mathbf{K} = 3$) generated by the IR Spectra Translator, IR-Agent achieves higher Top-$K$ accuracy compared to the standalone Transformer model, which functions as the IR Spectra Translator in our system. This improvement is attributed to the complementary insights provided by both experts. Their collaborative analyses enable the Structure Elucidation (SE) Expert to refine the candidates and generate more accurate final structures. (**2**) The multi-agent framework consistently outperforms the single-agent approach. Compared to the single-agent version of IR-Agent, in which a single LLM handles all

Table 1: Overall model performance for structure elucidation from IR spectra.

| Method | Agent | Top-K Accuracy | | | |
|---|---|---|---|---|---|
| | | Top-1 | Top-3 | Top-5 | Top-10 |
| Transformer | - | 0.098 (0.007) | 0.169 (0.000) | 0.176 (0.003) | 0.176 (0.003) |
| IR-Agent (GPT-4o-mini) | single | 0.072 (0.008) | 0.118 (0.002) | 0.133 (0.002) | 0.157 (0.003) |
| | multi | 0.093 (0.003) | 0.152 (0.003) | 0.167 (0.005) | 0.176 (0.005) |
| IR-Agent (GPT-4o) | single | 0.083 (0.004) | 0.135 (0.002) | 0.165 (0.007) | 0.194 (0.008) |
| | multi | 0.093 (0.007) | 0.153 (0.005) | 0.177 (0.005) | 0.204 (0.005) |
| IR-Agent (o3-mini) | single | 0.087 (0.006) | 0.153 (0.005) | 0.179 (0.002) | 0.197 (0.004) |
| | multi | **0.103** (0.005) | **0.178** (0.007) | **0.199** (0.004) | **0.216** (0.001) |

tasks simultaneously, the multi-agent version, where each expert agent is responsible for a specific sub-task, demonstrates more consistent and superior performance in the structure elucidation task. This observation is further supported by the following finding: While using a more advanced LLM backbone generally leads to improved performance, the multi-agent version of IR-Agent (GPT-4o)

---

[1] https://chem.libretexts.org/Ancillary_Materials/Reference/Reference_Tables/Spectroscopic_Reference_Tables/Infrared_Spectroscopy_Absorption_Table

Table 2: Overall model performance with various chemical information.

| Chemical Information | o3-mini | | | | IR-Agent (single) (o3-mini) | | | | IR-Agent (multi) (o3-mini) | | | |
|---|---|---|---|---|---|---|---|---|---|---|---|---|
| | Top-1 | Top-3 | Top-5 | Top-10 | Top-1 | Top-3 | Top-5 | Top-10 | Top-1 | Top-3 | Top-5 | Top-10 |
| No Knowledge | 0.073 | 0.131 | 0.157 | 0.185 | 0.087 | 0.153 | 0.179 | 0.197 | 0.103 | 0.178 | 0.199 | 0.216 |
| | (0.010) | (0.011) | (0.011) | (0.005) | (0.010) | (0.011) | (0.011) | (0.005) | (0.005) | (0.007) | (0.004) | (0.001) |
| Scaffold | 0.096 | 0.160 | 0.177 | 0.198 | 0.112 | 0.195 | 0.208 | 0.228 | 0.118 | 0.208 | 0.232 | 0.258 |
| | (0.002) | (0.006) | (0.003) | (0.003) | (0.003) | (0.009) | (0.008) | (0.010) | (0.003) | (0.009) | (0.008) | (0.010) |
| Carbon Count | **0.105** | 0.158 | 0.186 | 0.214 | 0.121 | 0.177 | 0.194 | 0.219 | 0.123 | 0.190 | 0.215 | 0.252 |
| | (0.009) | (0.014) | (0.014) | (0.013) | (0.003) | (0.008) | (0.010) | (0.009) | (0.003) | (0.0005) | (0.009) | (0.007) |
| Atom Types | 0.104 | **0.182** | **0.209** | **0.237** | **0.123** | **0.208** | **0.235** | **0.266** | **0.127** | **0.213** | **0.250** | **0.278** |
| | (0.011) | (0.007) | (0.005) | (0.003) | (0.006) | (0.003) | (0.011) | (0.009) | (0.006) | (0.003) | (0.011) | (0.009) |

achieves comparable or even superior accuracy compared to the single-agent system built on o3-mini, despite the former relying on a simpler model. This trend is also observed when comparing the multi-agent system (GPT-4o-mini) with the single-agent version (GPT-4o). This result underscores the effectiveness of the multi-agent framework in handling the molecular structure elucidation, demonstrating its strength in performing integrative analysis beyond what a single-agent can achieve. We further provide comparisons with other baselines in the Appendix C.2.

**Structure Elucidation with Chemical Information.** It is worth noting that IR-Agent is primarily developed with the assumption that the IR spectrum is the only available information; however, in practice, supplementary analyses often provide additional chemical data that can be leveraged to support structure elucidation (Alberts et al., 2024b). To reflect this practical scenario, we consider three types of chemical information: atom types, scaffold (i.e., molecular backbone), and carbon count. In each case, the relevant chemical information is appended as a textual sentence to the prompt of the corresponding expert agent as described in Section 4.4. From Table 2, we make the following observations: **(1)** Even a brief textual prompt containing chemical information can enhance the model's ability to predict accurate molecular structures. Without requiring any architecture modifications or retraining, IR-Agent is able to successfully incorporate additional information into each expert's reasoning process through prompt-based interaction, leveraging the inherent reasoning capabilities of LLMs. **(2)** Among the various types of chemical information, we observe that incorporating **Atom Types** information enables the model to generate more accurate molecular structures compared to other types of chemical information. This is due to the inherent challenge of determining the exact set of constituent elements solely from an IR spectrum. **(3)** However, incorporating any form of chemical information consistently improved performance over using only IR spectra (**No Knowledge**), highlighting the importance of a flexible framework capable of integrating various types of available chemical information. In conclusion, each expert in IR-Agent plays a distinct and effective role in structure elucidation while flexibly integrating various forms of chemical information, demonstrating the potential extensibility of the multi-agent framework for spectral analysis tasks. Further experiments evaluating the robustness of IR-Agent to ambiguous chemical information are provided in Appendix C.4.

## 5.3 IN-DEPTH ANALYSIS

**Ablation Studies.** To assess the contribution of each expert agent, we conduct ablation studies by selectively removing them from the system. As shown in Table 3, relying solely on the IR Spectra Translator without any expert assistance (**No Experts**) results in a significant drop in performance. Furthermore, using only one expert (i.e., either the **TI Expert only** or **Ret Expert only**) underperforms compared to the case where both experts are employed. When only the TI Expert is used, the system struggles to capture global structural patterns, whereas the Ret Expert alone often fails to extract fine-grained local information from the IR Absorption Table. Nevertheless, the Ret Ex-

Table 3: Ablation study of IR-Agent (o3-mini).

| Expert | Top-K Accuracy | | | |
|---|---|---|---|---|
| | Top-1 | Top-3 | Top-5 | Top-10 |
| No Expert | 0.073 | 0.131 | 0.157 | 0.185 |
| | (0.010) | (0.011) | (0.011) | (0.005) |
| TI Expert only | 0.089 | 0.154 | 0.171 | 0.190 |
| | (0.011) | (0.004) | (0.002) | (0.002) |
| Ret Expert only | 0.098 | 0.169 | 0.188 | 0.211 |
| | (0.003) | (0.006) | (0.001) | (0.003) |
| IR-Agent (TI + Ret) | **0.103** | **0.178** | **0.199** | **0.216** |
| | (0.005) | (0.007) | (0.004) | (0.001) |

pert alone achieves slightly better performance than the TI Expert alone, as it can access a broader range of structural patterns by leveraging multiple retrieved SMILES candidates, resulting in richer overall structural information. These results highlight that utilizing both TI and Ret Experts is essential for providing complementary structural insights, enabling effective integrative analysis for structure elucidation.

**Sensitivity Analysis: Number of SMILES candidates $\mathcal{C}$.** Moreover, we investigate how varying the number of SMILES candidates $\mathcal{C}$ affects performance. As shown in Figure 2 (a), the performance of IR-Agent improves as $\mathcal{C}$ increase up to 3 or 5, but tends to decline beyond that point. This degradation may be attributed to the introduction of noisy candidates, which can hinder the experts' reasoning. In particular, the TI Expert is required to manually align information from the IR absorption table with an increasing number of candidates,

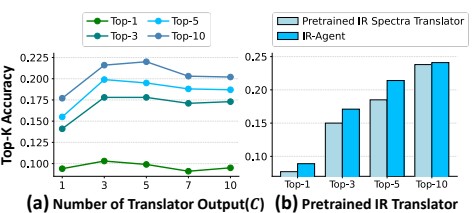

(a) Number of Translator Output($\mathcal{C}$)  (b) Pretrained IR Translator

Figure 2: In-depth Analysis results.

which raises the risk of incorporating irrelevant or misleading structural features. These results suggest that selecting an appropriate number of SMILES candidates is crucial for effective expert reasoning.

**Performance of IR-Agent using the Pretrained IR Spectra Translator.** As our framework is compatible with various IR spectra translators for generating initial SMILES candidates, we replace our original translator with a pretrained IR spectra translator trained on large-scale simulated data(Alberts et al., 2024a). This translator adopts a transformer architecture but differs from our original translator in its input representation: each IR spectrum is downsampled to 400 points, and the intensities are discretized into the range 0–99, enabling text-based tokenization of spectral values. Although simulated and experimental spectra differ in nature, the pretrained translator captures rich spectral patterns from large-scale simulated data, which remain beneficial when adapted to experimental data through fine-tuning. Figure 2 (b) shows that the pretrained translator achieves strong standalone performance, while its integration into our framework yields additional improvements, highlighting the robustness of our method across different spectra translator choices.

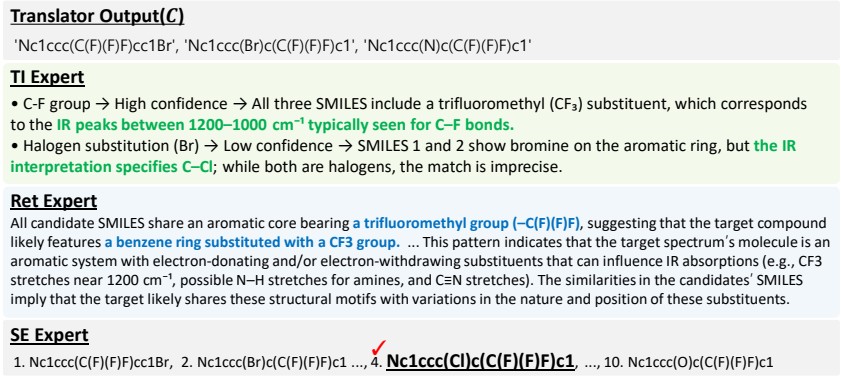

Figure 3: Outputs of expert agents in IR-Agent during the structure elucidation process.

**Case Study: How IR-Agent performs the structure elucidation.** In Figure 3, we present how IR-Agent performs the structure elucidation process. The TI Expert identifies local substructures by comparing the IR absorption table interpretation with the output of the IR Spectra Translator. For instance, the TI Expert infers the presence of a C–F group with high confidence, as both the Translator output and the table interpretation consistently point to C–F bonds. In contrast, the presence of a halogen substitution such as Br is inferred with low confidence, since the table interpretation refers to C–Cl bonds, which are not found in the Translator output. On the other hand, the Ret Expert extracts global structural patterns from the retrieved candidates, identifying a benzene ring substituted with a $CF_3$ group as a dominant motif, which serves as the broader structural context. Based on the complementary analyses from both experts, the SE Expert successfully infers the complete molecular structure of the target spectrum.

**Case Study: Structured reasoning by a Table Interpretation(TI) Expert.** In Figure 4, we demonstrate the structured reasoning process by which the TI expert identifies relevant local substructures from IR absorption table interpretations, rather than simply adopting table-based mappings. Although the two detected peak wavenumbers overlap with multiple wavenumber regions and initially correspond to six candidate substructures, the TI expert cross-validates these assignments against the translator outputs. As a result, only the isothiocyanate group associated with the 2140–1990 $\text{cm}^{-1}$

**Peaks wavenumber**
[1350.285. 2060.1487]

**Table interpretation mapping from the IR absorption table**
Peak near (1372, 1290) cm$^{-1}$ is associated with nitro compound (N-O).
Peak near (1420, 1330) cm$^{-1}$ is associated with alcohol (O-H).
Peak near (1390, 1310) cm$^{-1}$ is associated with phenol (O-H).
Peak near (1372, 1335) cm$^{-1}$ is associated with sulfonate (S=O).
Peak near (1370, 1335) cm$^{-1}$ is associated with sulfonamide (S=O).
Peak near (2140, 1990) cm$^{-1}$ is associated with isothiocyanate (N=C=S). ✓

**Translator Output($C$)**
'CC(C)c1ccc(C#N)cc1', 'CC(C)c1ccc(CN=C=S)cc1', 'C1=CCC1'

**TI Expert Reasoning**
- Isothiocyanate → high confidence → "CC(C)c1ccc(CN=C=S)cc1" **exhibits an N=C=S group, matching the IR peak between 2140–1990 cm$^{-1}$. (No IR-assigned substructures are found in "CC(C)c1ccc(C#N)cc1" or "C1=CCC1" based on the provided mapping.)**

**Answer SMILES**
'Cc1ccc(CCN=C=S)cc1'

Figure 4: Example of structured analytical reasoning by the Table Interpretation (TI) expert.

region is retained, while the remaining substructures are excluded because they do not appear in the translator outputs. This conclusion is consistent with the ground-truth SMILES, which also contains an isothiocyanate group.

# 6 CONCLUSION

In this paper, we propose IR-Agent , a novel multi-agent framework for structure elucidation from IR spectra that mimics the expert analytical process. To achieve this, we design a system composed of three specialized agents: a Table Interpretation (TI) Expert that extracts local substructures from the IR absorption table; a Retriever (Ret) Expert that provides global structural cues from retrieved candidates; and a Structure Elucidation (SE) Expert that integrates both sources of information to infer the final molecular structure. Furthermore, our framework supports the integration of chemical information in a lightweight and concise manner, which highlights its practicality in real-world analytical scenarios. Through extensive experiments including diverse chemical information conditions, we demonstrate both the effectiveness of IR-Agent in structure elucidation and the flexibility of the multi-agent framework in adapting to various analytical scenarios.

## ACKNOWLEDGEMENTS

This work was supported by grants from the National Research Foundation of Korea (NRF), funded by the Ministry of Science and ICT (RS-2022-NR068758 and RS-2024-00335098), and by the Institute of Information & Communications Technology Planning & Evaluation (IITP), funded by the Korea government (MSIT) (RS-2025-02304967, AI Star Fellowship (KAIST)).

## ETHICS STATEMENT

In this work, we present IR-Agent, a multi-agent system that automates molecular structure elucidation from IR spectra by emulating expert analytical processes. The reasoning capabilities of individual agents, each equipped with distinct knowledge, contribute to a more effective structure elucidation pipeline, with clearly defined and specialized reasoning steps. Unlike prior methods, the proposed multi-agent framework offers the flexibility to incorporate various types of chemical information, making it adaptable to a wide range of analytical scenarios. While our approach demonstrates promising performance in structure elucidation, it relies on large language model (LLM) agents, which may occasionally produce hallucinated outputs or misinterpret information. Therefore, it is essential that this framework be used under the supervision of domain experts when applied to real-world IR spectral analysis.

## REPRODUCIBILITY STATEMENT

To facilitate reproduction of the experimental results, we include all details in the main paper and appendix, describe the computational resources in Appendix A, and provide the accompanying code at `https://github.com/HeewoongNoh/IR-Agent`.

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

# Supplementary Material

## -IR-Agent: Expert-Inspired LLM Agents for Structure Elucidation from Infrared Spectra -

# A   IMPLEMENTATION DETAILS

Implementation details of IR-Agent are presented in this section.

## A.1   IR SPECTRA PREPROCESSING

We convert spectra from transmittance to absorbance using the standard formula:

$$A = -\log_{10}(T) \tag{6}$$

where $T$ denotes the transmittance and $A$ denotes the absorbance. To avoid mathematical errors during the logarithmic transformation and ensure numerical stability, any zero-valued entries are replaced with a small positive constant, $10^{-10}$.

## A.2   IR SPECTRA TRANSLATOR

**Analysis Setting Details.** While some studies (Alberts et al., 2024a; Wu et al., 2025) assume that the ground-truth chemical formula is always provided together with the IR spectrum, our framework relies solely on the IR spectrum. This assumption holds only if mass spectrometry (MS), performed alongside IR spectroscopy, can provide an accurate chemical formula. In reality, however, obtaining a ground-truth chemical formula is rarely straightforward, owing to several obstacles:

- Cost and Time: Generating high-quality MS data demands labor-intensive, time-consuming, and often expensive sample preparation (Vas & Vekey, 2004).
- Interpretation Complexity: MS spectra are notoriously challenging to interpret because of complex fragmentation patterns, overlapping peaks, and intrinsic resolution limits of the instrument (Rolland & Prell, 2021).
- Non-Trivial Formula Derivation: Even with high-quality spectra, determining the exact formula remains difficult. For example, studies on benchmark datasets such as NPLIB1—explicitly designed for this task—report Top-1 accuracies of only 48% (Böcker & Dührkop, 2016) and 71% (Goldman et al., 2023), underscoring the difficulty of precise formula determination.

Given these challenges, it is impractical to assume that the ground-truth chemical formula is always available for an unknown material. Accordingly, we adopt a setting where the chemical formula is not used by default (Ellis et al., 2023), and instead employ a SMILES translator that predicts molecular structures directly from IR spectra without formula information.

**Model Implementation Details.** We represent the IR spectrum as a 1D sequence $\mathcal{X} \in \mathbb{R}^{1 \times L}$, where $L$ is the number of absorbance values aligned with wavenumber positions. This input is then passed through a learnable linear transformation to produce a higher-dimensional feature sequence $\mathbf{x} \in \mathbb{R}^{L \times d}$. To inject positional information, we define a learnable positional embedding matrix $\mathbf{P} \in \mathbb{R}^{L \times d}$, where each row $\mathbf{P}_i$ corresponds to the positional embedding at the $i$-th wavenumber. The input to the Transformer encoder is computed as:

$$\mathbf{z}_i = \mathbf{x}_i + \mathbf{P}_i, \quad \text{for } i = 1, \dots, L. \tag{7}$$

The resulting spectrum representations $\mathbf{Z} = \{\mathbf{z}_i\}_{i=1}^{L} \in \mathbb{R}^{L \times d}$ are first fed into the Transformer encoder and subsequently into the Transformer decoder, which autoregressively generates the target molecular sequence. The model is trained to maximize the likelihood of the ground-truth output tokens given the input spectrum by minimizing the following cross-entropy (CE) loss:

$$\mathcal{L}_{\text{CE}} = -\frac{1}{N} \sum_{n=1}^{N} \sum_{t=1}^{T_n} \log p_\theta(y_t^{(n)} \mid y_{<t}^{(n)}, \mathcal{X}^{(n)}), \tag{8}$$

where $N$ is the number of training examples, $T_n$ is the length of the target sequence for the $n$-th example, $y_t^{(n)}$ denotes the $t$-th token in the ground-truth molecular SMILES for the $n$-th example, and $p_\theta$ denotes the model's predicted probability distribution parameterized by $\theta$, where $\theta$ represents the learnable parameters of the translator. This objective corresponds to the standard next-token prediction loss widely used in training language models (Vaswani et al., 2017).

**Training Details.** The Translator is implemented in Python 3.11.10 and PyTorch 2.5.1. We use the Adam optimizer for model training. The model is trained for up to 300 epochs, with early stopping applied if the best validation BLEU score does not improve for 25 consecutive epochs. All the experiments are conducted on a 48GB NVIDIA RTX A6000.

**Hyperparameters.** For the Translator, we use a batch size of 16, hidden dimension $d = 128$, and a learning rate of 0.001 with a linear scheduler and 8,000 warm-up steps. The model consists of 2 encoder layers and 2 decoder layers, and the number of retrieved spectra ($N$) is set to 10. A beam width of 3 is chosen to reflect a practical decoding setting with moderate computational cost. A comparison of performance across larger beam widths is provided in Section C.1. Note that when the model fails to generate the desired number of outputs, we apply greedy decoding to supplement the remaining outputs and guarantee a fixed output size. We perform a grid search over learning rates {0.0001, 0.0005, 0.001}, batch sizes {16, 32, 64}, and hidden dimensions {64, 128}, and report test performance based on the best model selected according to validation set results.

### A.3    IR ABSORPTION TABLE

Table 4 shows the IR absorption table used in this paper, which is available online [2]. For wavenumber entries specified as single points rather than ranges, we convert them into ranges by applying a $\pm 5 \text{ cm}^{-1}$ window around each point.

Table 4: Wavenumber Range and Substructure Assignments

| Wavenumber ($\text{cm}^{-1}$) | Substructure |
|---|---|
| 3700–3584 | alcohol (O–H) |
| 3550–3200 | alcohol (O–H) |
| 3500–3400 | primary amine (N–H) |
| 3400–3300 | aliphatic primary amine (N–H) |
| 3330–3250 | aliphatic primary amine (N–H) |
| 3350–3310 | secondary amine (N–H) |
| 3100–2900 | carboxylic acid (O–H) |
| 3200–2700 | alcohol (O–H) |
| 3000–2800 | amine salt (N–H) |
| 3333–3267 | alkyne (C–H) |
| 3100–3000 | alkene (C–H) |
| 3080–2840 | alkane (C–H) |
| 2830–2695 | aldehyde (C–H) |
| 2600–2550 | thiol (S–H) |
| 2354–2344 | carbon dioxide (O=C=O) |
| 2285–2250 | isocyanate (N=C=O) |
| 2260–2222 | nitrile (C≡N) |
| 2260–2190 | disubstituted alkyne (C≡C) |
| 2175–2140 | thiocyanate (S–C≡N) |
| 2160–2120 | azide (N=N=N) |
| 2155–2145 | ketene (C=C=O) |
| 2145–2120 | carbodiimide (N=C=N) |
| 2140–2100 | monosubstituted alkyne (C≡C) |
| 2140–1990 | isothiocyanate (N=C=S) |
| 2005–1995 | ketenimine (C=C=N) |
| 2000–1900 | allene (C=C=C) |
| 2000–1650 | aromatic compound (C–H) |
| 1818–1750 | anhydride (C=O) |
| 1815–1785 | acid halide (C=O) |
| 1800–1770 | conjugated acid halide (C=O) |

Continued on next page

---

[2]https://chem.libretexts.org/Ancillary_Materials/Reference/Reference_ Tables/Spectroscopic_Reference_Tables/Infrared_Spectroscopy_Absorption_ Table

Table 4: Wavenumber Range and Substructure Assignments

| Wavenumber (cm$^{-1}$) | Substructure |
| --- | --- |
| 1780–1770 | conjugated anhydride (C=O) |
| 1770–1780 | vinyl/phenyl ester (C=O) |
| 1765–1755 | carboxylic acid (C=O) |
| 1750–1735 | esters (C=O) |
| 1750–1740 | cyclopentanone (C=O) |
| 1740–1720 | aldehyde (C=O) |
| 1730–1715 | α,β–unsaturated ester (C=O) |
| 1725–1715 | conjugated anhydride (C=O) |
| 1725–1705 | aliphatic ketone (C=O) |
| 1720–1706 | carboxylic acid (C=O) |
| 1710–1685 | conjugated aldehyde (C=O) |
| 1710–1680 | conjugated acid (C=O) |
| 1695–1685 | primary amide (C=O) |
| 1690–1640 | imine/oxime (C=N) |
| 1685–1675 | tertiary amide (C=O) |
| 1685–1666 | conjugated ketone (C=O) |
| 1678–1668 | trans–disubstituted alkene (C=C) |
| 1675–1665 | tetrasubstituted alkene (C=C) |
| 1662–1626 | cis–disubstituted alkene (C=C) |
| 1658–1600 | alkene (vinylidene) (C=C) |
| 1655–1645 | δ–lactam (C=O) |
| 1650–1600 | conjugated alkene (C=C) |
| 1650–1580 | amine (N–H) |
| 1650–1566 | cyclic alkene (C=C) |
| 1648–1638 | monosubstituted alkene (C=C) |
| 1620–1610 | α,β–unsaturated ketone (C=C) |
| 1550–1500 | nitro compound (N–O) |
| 1470–1460 | alkane (methylene group) (C–H) |
| 1455–1445 | alkane (methyl group) (C–H) |
| 1440–1395 | carboxylic acid (O–H) |
| 1420–1330 | alcohol (O–H) |
| 1415–1380 | sulfate (S=O) |
| 1410–1380 | sulfonyl chloride (S=O) |
| 1390–1380 | aldehyde (C–H) |
| 1390–1310 | phenol (O–H) |
| 1385–1380 | alkane (gem dimethyl) (C–H) |
| 1380–1370 | alkane (methyl group) (C–H) |
| 1372–1335 | sulfonate (S=O) |
| 1372–1290 | nitro compound (N–O) |
| 1370–1365 | alkane (gem dimethyl) (C–H) |
| 1370–1335 | sulfonamide (S=O) |
| 1350–1342 | sulfonic acid (S=O) |
| 1350–1300 | sulfone (S=O) |
| 1342–1266 | aromatic amine (C–N) |
| 1310–1250 | aromatic ester (C–O) |
| 1300–1250 | phosphorus oxide (P–O) |
| 1275–1200 | alkyl aryl ether (C–O) |
| 1250–1195 | phosphorus oxide (P–O) |
| 1250–1020 | amine (C–N) |
| 1225–1200 | vinyl ether (C–O) |
| 1210–1163 | ester (C–O) |
| 1205–1124 | tertiary alcohol (C–O) |
| 1204–1177 | sulfonyl chloride (S=O) |
| 1200–1185 | sulfate (S=O) |
| 1200–1000 | fluoro compound (C–F) |

Continued on next page

Table 4: Wavenumber Range and Substructure Assignments

| Wavenumber (cm$^{-1}$) | Substructure |
| --- | --- |
| 1195–1168 | sulfonate (S=O) |
| 1170–1155 | sulfonamide (S=O) |
| 1165–1150 | sulfonic acid (S=O) |
| 1160–1120 | sulfone (S=O) |
| 1150–1085 | aliphatic ether (C–O) |
| 1124–1087 | secondary alcohol (C–O) |
| 1085–1050 | primary alcohol (C–O) |
| 1075–1020 | alkyl aryl ether (C–O) |
| 1075–1020 | vinyl ether (C–O) |
| 1070–1030 | sulfoxide (S=O) |
| 1050–1040 | anhydride (CO–O–CO) |
| 995–985 | monosubstituted alkene (C=C) |
| 915–905 | monosubstituted alkene (C=C) |
| 980–960 | trans–disubstituted alkene (C=C) |
| 895–885 | alkene (vinylidene) (C=C) |
| 840–790 | trisubstituted alkene (C=C) |
| 760–540 | halo compound (C–Cl) |
| 730–665 | cis–disubstituted alkene (C=C) |
| 690–515 | halo compound (C–Br) |
| 600–500 | halo compound (C–I) |
| 750–700 | monosubstituted benzene derivative |
| 710–690 | monosubstituted benzene derivative |

### A.4 IR Peak Table Assigner

The IR Peak Table Assigner consists of two main components: (1) extracting peaks from the input spectrum, and (2) identifying relevant substructures using the IR Absorption Table (Table 4). To extract peaks, we use the `find_peaks` function from SciPy[3], setting the hyperparameters to `height=1` and `distance=50`. This allows us to identify wavenumber positions where the absorbance exhibits local maxima, which we refer to as peaks. After extracting the peaks, we assign the corresponding substructures based on the IR absorption table, and generate textual interpretations. For example, if a peak is found within the range of (1200, 1000) cm$^{-1}$, the interpretation might be: "Peaks observed between 1200 and 1000 cm$^{-1}$ are typically associated with fluoro compounds (C–F)."

### A.5 LLM Agents

**System Setup.** Our agent system is implemented using Python 3.11.10, with `langchain` 0.3.25, `langchain-openai` 0.2.11, and `langgraph` 0.2.59. We utilize three LLMs: GPT-4o-mini[4], GPT-4o[5], and o3-mini[6]. For GPT-4o-mini, we use the `gpt-4o-mini-2024-07-18` model with a temperature setting of 0.8. For GPT-4o, we use `gpt-4o-2024-08-06`, also with a temperature of 0.8. For o3-mini, we use `o3-mini-2025-01-31` with default settings and medium reasoning mode.

**Computational Cost Analysis.** In Table 5, we summarize the cost per LLM, along with input and output token counts. Although the output length for both the TI and Ret Experts is constrained to fewer than 300 tokens, we observe that `o3-mini`, which is designed as a reasoning model, tends to generate a higher number of output tokens due to reasoning tokens that are not explicitly reflected in the final output. The overall API cost for a single run of IR-Agent increases in the order of GPT-4o-mini, GPT-4o, and o3-mini. Additionally, we find that LLMs with more intricate reasoning processes

---

[3] `https://docs.scipy.org/doc/scipy/reference/generated/scipy.signal.find_peaks.html`
[4] `https://openai.com/index/gpt-4o-mini-advancing-cost-efficient-intelligence/`
[5] `https://openai.com/index/hello-gpt-4o/`
[6] `https://openai.com/index/openai-o3-mini/`

Table 5: Computational cost analysis comparison across different LLMs.

| Computational cost & Tokens | GPT-4o-mini | GPT-4o | o3-mini |
|---|---|---|---|
| Input cost (per 1M token) | $0.15 | $ 2.50 | $1.10 |
| Output cost (per 1M token) | $0.6 | $10.00 | $4.40 |
| Average input token | 1500 | 1500 | 1400 |
| Average output token | 600 | 600 | 4400 |
| Avg. cost per call (IR-Agent) | $0.0006 | $0.0097 | $0.0209 |
| TI Expert (sec) | 3.6 | 5.6 | 16.2 |
| Ret Expert (sec) | 6.2 | 5.5 | 8.2 |
| SE Expert (sec) | 3.2 | 2.7 | 18 |
| IR-Agent(single, sec) | 3.9 | 2.8 | 13.5 |

generally exhibit longer average runtimes per sample. Since the TI and Ret Experts can be operated in parallel, the total runtime for IR-Agent is determined by whichever of these two is slower, along with the time required for the SE Expert. The IR-Agent (single), which performs all tasks at once, is much faster than the multi-agent approach. However, its performance is lower compared to the multi-agent setup, indicating a trade-off between speed and accuracy. The complete process—which involves interpreting the IR absorption table, extracting key features from retrieved SMILES, and leveraging this information to generate the final SMILES candidates—naturally requires a considerable amount of time. While the current LLM inference speed is not sufficient for high-throughput or large-scale applications, IR-Agent is still faster than manual expert analysis. Thus, it can serve as an effective tool for offering structural suggestions before a human expert undertakes detailed spectrum interpretation.

## B  DATASET

**Preprocessing Details.** Unlike prior work (Alberts et al., 2024a; Wu et al., 2025), we do not exclude compounds with stereochemistry or ionic states, nor do we restrict the heavy atom count to between 6 and 13 or limit elemental composition to C, H, N, O, S, P, and halogens. Instead, the NIST dataset we use contains 9,052 spectra with diverse phase compositions—56% gas, 20% liquid, and 24% solid—capturing broader chemical diversity. The heavy atom count ranges from 3 to 68 (mean: 13.4, median: 12.0), which is substantially higher and more variable than in previous datasets. No filtering is imposed based on stereochemistry, charge state, or elemental composition; all spectra are retained. Consequently, our dataset exhibits higher SMILES token diversity and better reflects real-world experimental conditions. Following Na (2024), we apply polynomial interpolation over wavenumbers ranging from $500$–$4000$ cm$^{-1}$ to obtain a structured format, addressing the inconsistent number of absorbance points across samples. For spectra recorded in transmittance mode, intensity values are converted to absorbance using a standard conversion A.1.

## C  FURTHER ANALYSIS

### C.1  PERFORMANCE COMPARISON ACROSS DIFFERENT BEAM WIDTHS

We validate the effectiveness of IR-Agent when using the IR Spectra Translator with larger beam widths. As shown in Figure 5, increasing the beam width beyond the default setting of 3 improves the performance of the Translator, thanks to the increased diversity in the decoding process. Moreover, when integrated into our framework, IR-Agent consistently yields additional performance gains by leveraging the SMILES candidates generated by the enhanced translator.

### C.2  PERFORMANCE COMPARISON WITH OTHER BASELINES

Table 6: Comparison with Additional Baselines

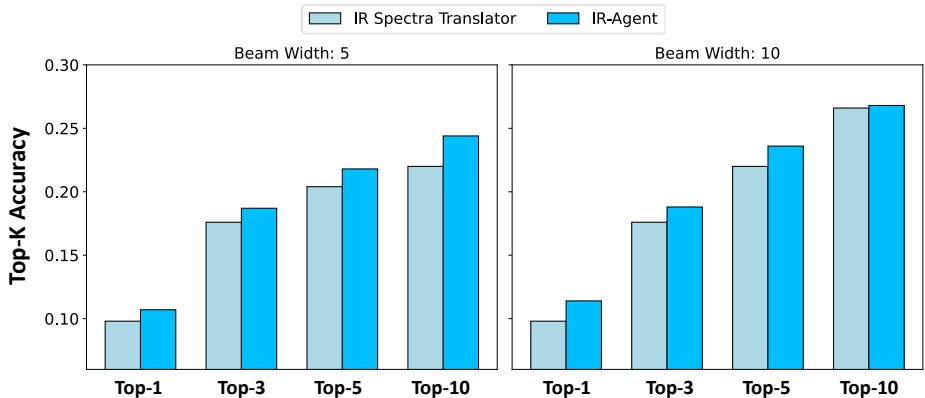

Figure 5: Performance of IR-Agent using Translator across different beam widths

| Model | Top-K Accuracy | | | | Tanimoto Similarity | | |
|---|---|---|---|---|---|---|---|
| | Top-1 | Top-3 | Top-5 | Top-10 | MACCS | RDKit | Morgan |
| LLaMA-3.1-8B(Grattafiori et al., 2024) | - | - | - | - | - | - | - |
| Patch-Based Self-Attention Transformer (Wu et al., 2025) | **0.129** (0.003) | 0.160 (0.002) | 0.170 (0.003) | 0.181 (0.004) | 0.721 (0.006) | 0.547 (0.008) | 0.518 (0.007) |
| IR-Agent | 0.103 (0.005) | **0.178** (0.007) | **0.199** (0.004) | **0.216** (0.001) | **0.770** (0.004) | **0.596** (0.002) | **0.549** (0.003) |

We conduct additional experiments with two baselines: (1) Llama-3-8B(Grattafiori et al., 2024), and (2) the Patch-Based Self-Attention Transformer(Wu et al., 2025). Since IR spectra consist of thousands of floating-point intensity values, we downsample each spectrum to 400 points and normalize the intensities to the range 0–99 to make tokenization feasible. Using this representation, we evaluate Llama-3-8B under two settings: (1) zero-shot inference with the pretrained model and (2) QLoRA-based fine-tuning.

In both settings, the model fails to generate valid SMILES strings and often produced severely corrupted or nonsensical outputs. For the zero-shot case, this outcome is expected because the model has never been exposed to any form of mapping between IR values and molecular structures. For the fine-tuning case, the difficulty appears to stem from the relatively modest dataset size and the challenge of learning meaningful spectral patterns directly from numeric sequences. The downsampling step may also have introduced information loss, further degrading model performance. An example of the data used in the Llama experiments is provided below.

['role': 'system', 'content': 'You are an expert for IR spectroscopy, especially for molecular structure elucidation', 'role': 'user', 'content': 'Given IR absorbance: IR 0 2 3 5 6 8 9 11 12 14 15 17 18 20 21 23 24 26 27 29 30 32 33... 46 46, generate the corresponding SMILES.']

In addition to the Llama experiments, we also compare IR-Agent with another Transformer architecture featuring an advanced attention mechanism. The Patch-Based Self-Attention Transformer(Wu et al., 2025) represents the state-of-the-art non-agentic method for IR-based structure elucidation, incorporating a sophisticated patch-level attention design along with two data-augmentation strategies. To ensure a fair comparison, we used the official GitHub implementation and evaluated the model on the same test dataset as IR-Agent. Our results show that although the Patch-Based Self-Attention model attains higher Top-1 accuracy, IR-Agent outperforms it across all other exact-match metrics as well as all Tanimoto similarity measures. This indicates that our approach produces structurally more meaningful and chemically relevant predictions.

## C.3 ROBUSTNESS OF LLMs TO PROMPT VARIATIONS

Table 7: Robustness of IR-Agent (o3-mini) to prompt variations.

| Prompt | Top-K Accuracy | | | |
|---|---|---|---|---|
| | Top-1 | Top-3 | Top-5 | Top-10 |
| Prompt 1 | **0.110** | 0.168 | 0.194 | 0.214 |
| | (0.003) | (0.005) | (0.007) | (0.005) |
| Prompt 2 | 0.100 | **0.182** | **0.201** | **0.222** |
| | (0.006) | (0.004) | (0.011) | (0.006) |
| IR-Agent | 0.103 | 0.178 | 0.199 | 0.216 |
| | (0.005) | (0.007) | (0.004) | (0.001) |

To evaluate the robustness of IR-Agent across different prompts, we curated new prompt variations for each agent by rephrasing the originals. Specifically, we asked GPT-4o to "rephrase the given prompt with the same semantics, but in a different structure," and used the resulting two paraphrased versions for additional experiments with IR-Agent. Notably, IR-Agent demonstrated robust performance across these entirely paraphrased prompts, with results showing little deviation from the standard deviation observed with the original prompts.

## C.4 ROBUSTNESS OF LLMS TO AMBIGUOUS CHEMICAL INFORMATION

Table 8: Robustness of IR-Agent (o3-mini) to ambiguous chemical information.

| Chemical Information | Top-K Accuracy | | | |
|---|---|---|---|---|
| | Top-1 | Top-3 | Top-5 | Top-10 |
| No Knowledge (table 2) | 0.103 | 0.178 | 0.199 | 0.216 |
| | (0.005) | (0.007) | (0.004) | (0.001) |
| Ambiguous Carbon Num | 0.091 | 0.188 | 0.213 | 0.249 |
| | (0.003) | (0.003) | (0.006) | (0.005) |
| Exact Carbon Num (table 2) | **0.123** | **0.190** | **0.215** | **0.252** |
| | (0.003) | (0.005) | (0.009) | (0.007) |

In practical experimental settings, chemical information is often ambiguous or incomplete. To reflect this, we also consider an "ambiguous carbon number" scenario, where the number of carbons is provided as a range (i.e., exact carbon number ±1) rather than an exact value. We evaluate the performance of IR-Agent under this setting with incomplete information. Experimental results show that, as expected, the performance of IR-Agent drops when given an ambiguous carbon number compared to the exact value. However, except for the Top-1 metric, the decrease in performance is not significant, indicating that IR-Agent can still accurately predict the correct SMILES even when only incomplete carbon information is available.

## C.5 DETERMINISTIC WORKFLOW OF IR-AGENT VS. THE REACT FRAMEWORK

Table 9: Comparison of IR-Agent (GPT-4o) with ReAcT framework

| Workflow | Top-K Accuracy | | | |
|---|---|---|---|---|
| | Top-1 | Top-3 | Top-5 | Top-10 |
| ReAcT Framework | 0.083 | 0.148 | 0.151 | 0.158 |
| | (0.004) | (0.005) | (0.003) | (0.005) |
| IR-Agent | **0.093** | **0.153** | **0.177** | **0.204** |
| | (0.007) | (0.005) | (0.005) | (0.005) |

IR-Agent follows a largely deterministic and fixed pattern, lacking the dynamic decision-making characteristic of agent systems. Since experts necessarily refer to both IR absorption tables and spectral databases when analyzing IR spectra, they rely on the combination of these two sources of information for interpretation. Unlike common tasks such as question answering, structure elucidation from IR spectra is highly specific and challenging, where faithfully emulating the expert reasoning process is essential. Therefore, we aim to design a system that closely mirrors the analytical workflow of domain experts. We conduct a comparative experiment using a ReAct agent framework, where the LLM autonomously selects its actions. Specifically, we employ GPT-4o as the base LLM and provided a set of SMILES candidates generated by the IR Spectra Translator. The available tools include the IR Peak Table Assigner, IR Spectra Retriever, and an additional finish tool that enables the agent to terminate tool selection and output the final SMILES. The maximum number of tool calls is limited to five. In line with the ReAct framework, the agent iteratively goes through thought, action (tool selection), and observation steps.

From our experimental results, we observe that IR-Agent demonstrates superior performance compared to the ReAct framework. In the ReAct setting, the agent often successfully calls both the IR Peak Table Assigner and the IR Spectra Retriever. However, we also found two common failure modes: (1) the agent selects only the IR Peak Table Assigner, limiting itself to a narrow set of information, and (2) the agent repeatedly selects the same tool (e.g., retriever, then table, then table again), resulting in final SMILES predictions that are biased toward the information provided by that tool alone. These observations suggest that, for the task of structure elucidation from IR spectra, which requires substantial domain knowledge and careful emulation of expert reasoning, a deterministic approach to tool selection—as implemented in IR-Agent—is necessary. Such a deterministic agent process is effective in emulating the expert analysis workflow and highlights the importance of guided, expert-like decision-making in this context

## C.6 THE SCOPE OF IR-AGENT

Table 10: Performance of IR-Agent on Single Compounds and Mixtures

| Type | # Test | Top-1 | Top-3 | Top-5 | Top-10 | MACCS | RDK | Morgan |
|------|--------|-------|-------|-------|--------|-------|-----|--------|
| Single | 886 | $0.105_{(0.006)}$ | $0.183_{(0.008)}$ | $0.204_{(0.005)}$ | $0.221_{(0.001)}$ | $0.775_{(0.003)}$ | $0.603_{(0.003)}$ | $0.556_{(0.003)}$ |
| Mixture | 20 | 0.000 | 0.000 | 0.000 | 0.000 | $0.567_{(0.026)}$ | $0.310_{(0.043)}$ | $0.277_{(0.027)}$ |

Unlike previous studies(Alberts et al., 2024a; Wu et al., 2025; Jung et al., 2023) that typically restrict their scope to single gas-phase materials, we aimed to design our model under a more realistic and unconstrained setting using the NIST dataset. Consequently, our dataset (total: 9,052 spectra) includes both single compounds (8,810) and mixtures (242). However, peak overlap in mixtures poses a significant challenge for accurate prediction. Our empirical results show that although IR-Agent performs exceptionally well on single compounds, it struggles with mixtures due to signal complexity and limited training examples. Therefore, our IR-Agent focuses on handling single molecular components, and we consider mixture analysis an important but challenging direction for future work to further enhance the agent's capabilities.

## C.7 ADDITIONAL CASE STUDIES

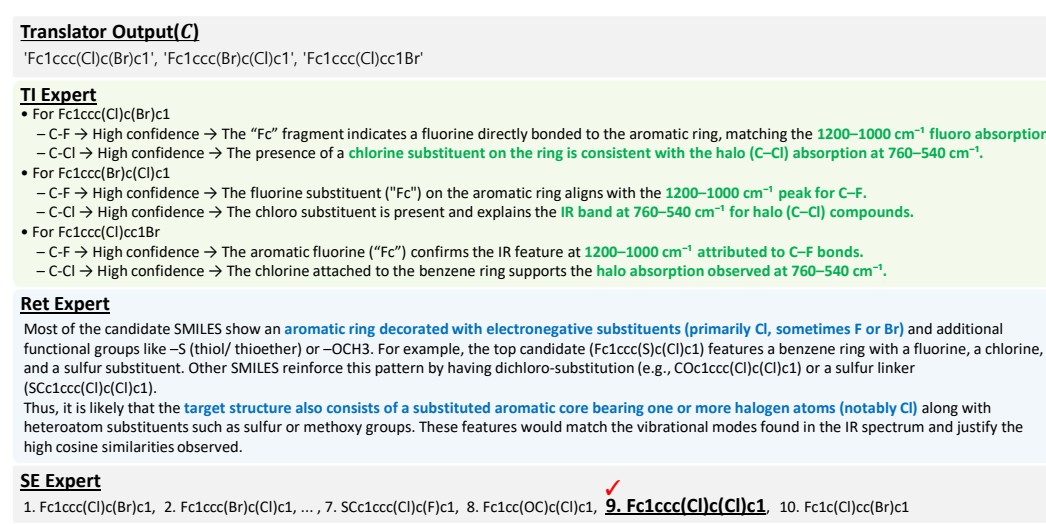

Figure 6: Additional Case Study: Outputs of expert agents in IR-Agent.

In Figure 6, we present an additional case study for IR-Agent. The TI Expert infers the presence of halogen atoms such as F and Cl with high confidence, based on the consistency between the translator output and the table interpretation (e.g., C–F, C–Cl bonds). Additionally, the Ret Expert identifies a structural pattern consisting of an aromatic ring substituted with one or more halogen atoms. Based on these integrative analyses, the SE Expert accurately infers the complete molecular structure of the target spectrum.

## D   LIMITATIONS & FUTURE WORK

We focus on extracting local structural information based on interpretations from the IR absorption table. However, accurate interpretation requires considering not only the peak positions, but also the peak shapes and intensities. Since our framework refines and regenerates SMILES based on the candidates provided by the IR Spectra Translator, its overall performance is naturally influenced by the Translator. At the same time, the framework is designed to flexibly incorporate diverse types of chemical information without requiring retraining or architectural modifications. Nevertheless, when adapting to new spectral datasets, the IR Spectra Translator itself still needs to be retrained, as is the case with prior approaches. An alternative approach would be to directly input the IR spectrum as an image along with its raw spectral values into an LLM. With an effective prompting strategy or collaboration with external tools, this would enable the model to capture peak shapes and intensities during the interpretation of the IR absorption table, and to generate candidate SMILES without requiring retraining when adapting to new spectral datasets, effectively functioning as a translator.

## E   PROMPT TEMPLATES FOR EXPERT AGENTS

In this section, we provide the prompt templates used for each agent described in Section 4. In addition to the default setting without chemical information, we present modified prompt templates that incorporate additional chemical information in three scenarios: atom types, scaffold, and carbon count. For each case, a single sentence describing the given chemical information is appended to the original prompt.

Table 11: Prompt for Table Interpretation (TI) Expert (Section 4.1)

---

**System Prompt:** You are an expert organic chemist with specialized knowledge in analyzing infrared (IR) spectra.

**Prompt:** You have an IR absorption interpretation that suggests certain substructures (e.g. nitrile, carbonyl, etc.), but this table-based mapping can be imprecise.

Given SMILES: {SMILES Candidates}
IR interpretation: {Table Interpretation}

Your task is to:
For each SMILES in the given SMILES list, identify substructures that are present both in the IR interpretation and in that SMILES.

Return a bulleted list in the format:
substructure → confidence → brief rationale

KEEP THE RESPONSE UNDER 300 TOKENS.
ONLY RETURN:
- A bulleted list of (substructure → confidence → brief rationale).

---

Table 12: Prompt for Table Interpretation (TI) Expert with Chemical Information(Section 4.4)

**System Prompt:** You are an expert organic chemist with specialized knowledge in analyzing infrared (IR) spectra.

**Prompt:** You have an IR absorption interpretation that suggests certain substructures (e.g. nitrile, carbonyl, etc.), but this table-based mapping can be imprecise.

Given SMILES: {SMILES Candidates}
IR interpretation: {Table Interpretation}

The molecule corresponding to the target spectrum is known to include the following atom types: **{Atom Types}**.
The molecule corresponding to the target spectrum is known to include the following scaffold: **{Scaffold}**.
The molecule corresponding to the target spectrum is known to include exactly **{Carbon Count}** carbon atoms.

Your task is to:
For each SMILES in the given SMILES list, identify substructures that are present both in the IR interpretation and in that SMILES.

Return a bulleted list in the format:
substructure → confidence → brief rationale

KEEP THE RESPONSE UNDER 300 TOKENS.
ONLY RETURN:
- A bulleted list of (substructure → confidence → brief rationale).

Table 13: Prompt for Retriever (Ret) Expert (Section 4.2)

**System Prompt:** You are an expert organic chemist with specialized knowledge in analyzing infrared (IR) spectra.

**Prompt:** Your task is to analyze the SMILES of the candidate spectra, whose cosine similarity to the target spectrum is high.

If the target spectrum and candidate spectra exhibit high similarity, the SMILES of the target spectrum may have a similar structural characteristics to the SMILES of the candidate spectrum.

SMILES of candidate spectra and their cosine similarities to the target spectrum:
{Output of IR Spectra Retriever}

Based on the SMILES list, extract the structural information to complement the SMILES of the target spectrum.

Provide reasoning to support your analysis.

Let's think step-by-step.

KEEP THE RESPONSE UNDER 300 TOKENS.

ONLY THE REQUESTED CONTENT SHOULD BE INCLUDED IN YOUR RESPONSE.

Table 14: Prompt for Retriever (Ret) Expert with Chemical Information (Section 4.4)

**System Prompt:** You are an expert organic chemist with specialized knowledge in analyzing infrared (IR) spectra.

**Prompt:** Your task is to analyze the SMILES of the candidate spectra, whose cosine similarity to the target spectrum is high.

If the target spectrum and candidate spectra exhibit high similarity, the SMILES of the target spectrum may have a similar structural characteristics to the SMILES of the candidate spectrum.

SMILES of candidate spectra and their cosine similarities to the target spectrum:
{Output of IR Spectra Retriever}

The molecule corresponding to the target spectrum is known to include the following atom types: **{Atom Types}**.
The molecule corresponding to the target spectrum is known to include the following scaffold: **{Scaffold}**.
The molecule corresponding to the target spectrum is known to include exactly
**{Carbon Count}** carbon atoms.

Based on the SMILES list, extract the structural information to complement the SMILES of the target spectrum.

Provide reasoning to support your analysis.

Let's think step-by-step.

KEEP THE RESPONSE UNDER 300 TOKENS.

ONLY THE REQUESTED CONTENT SHOULD BE INCLUDED IN YOUR RESPONSE.

Table 15: Prompt for Structure Elucidation (SE) Expert (Section 4.3)

---

**System Prompt:** You are an expert organic chemist with specialized knowledge in analyzing infrared (IR) spectra.

**Prompt:** Your task is to refine the given SMILES list and generate a N candidate list that aligns well with the IR spectrum while preserving structural diversity and plausibility.

The IR Absorption Table Agent provides potentially useful insights by interpreting the IR spectrum and suggesting possible substructures based on known absorption patterns.

IR Spectrum Retriever Agent examines the structural features of candidate SMILES that exhibit high cosine similarity to the target spectrum.

IR Absorption Table Agent Output: $\{\mathcal{A}_{\text{TI Expert}}\}$

IR Spectrum Retriever Agent Output (high-similarity spectra & analysis): $\{\mathcal{A}_{\text{Ret Expert}}\}$

1) Identify the substructures that are common to both the IR table interpretation and at least one SMILES in the list.

2) From the retriever agent output, extract structural information (e.g., recurring motifs / scaffolds) suggested by high-similarity candidates.

3) Guided by the structural insights from steps 1 and 2, produce a refined Top-N list of SMILES candidates.

4) Ensure the final list is chemically diverse and plausible—do not overfit to any single interpretation.

Based on these analyses, regenerate a list of Top-N SMILES by refining the target smiles: {SMILES Candidates}.

Let's think step-by-step.

ONLY THE REQUESTED CONTENT SHOULD BE INCLUDED IN YOUR RESPONSE.

YOUR ANSWER FORMAT MUST BE AS FOLLOWS ONLY CONTAINING THE SMILES:
1. SMILES_1, 2. SMILES_2, 3. SMILES_3, ..., N. SMILES_N

---

Table 16: Prompt for Structure Elucidation (SE) Expert with Chemical Information (Section 4.4)

**System Prompt:** You are an expert organic chemist with specialized knowledge in analyzing infrared (IR) spectra.

**Prompt:** Your task is to refine the given SMILES list and generate a N candidate list that aligns well with the IR spectrum while preserving structural diversity and plausibility.

The IR Absorption Table Agent provides potentially useful insights by interpreting the IR spectrum and suggesting possible substructures based on known absorption patterns.

IR Spectrum Retriever Agent examines the structural features of candidate SMILES that exhibit high cosine similarity to the target spectrum.

IR Absorption Table Agent Output: $\{\mathcal{A}_{\text{TI Expert}}\}$

IR Spectrum Retriever Agent Output (high-similarity spectra & analysis): $\{\mathcal{A}_{\text{Ret Expert}}\}$

The final predicted molecular structures are constrained to contain only the following atom types:**{Atom Types}**.
The final predicted molecular structures must incorporate the specified scaffold **{Scaffold}**.
The final predicted molecular structures are required to contain exactly **{Carbon Count}** carbon atoms.
1) Identify the substructures that are common to both the IR table interpretation and at least one SMILES in the list.

2) From the retriever agent output, extract structural information (e.g., recurring motifs / scaffolds) suggested by high-similarity candidates.

3) Guided by the structural insights from steps 1,2, and **[{Atom Types}, {Scaffold}, {Carbon Count}]** constraint, produce a refined Top-N list of SMILES candidates.

4) Ensure the final list is chemically diverse and plausible—do not overfit to any single interpretation.

Based on these analyses, regenerate a list of Top-N SMILES by refining the target smiles: {SMILES Candidates}.

Let's think step-by-step.

ONLY THE REQUESTED CONTENT SHOULD BE INCLUDED IN YOUR RESPONSE.

YOUR ANSWER FORMAT MUST BE AS FOLLOWS ONLY CONTAINING THE SMILES:
1. SMILES_1, 2. SMILES_2, 3. SMILES_3, ..., N. SMILES_N

