# OpenReview forum: "IR-Agent: Expert-Inspired LLM Agents for Structure Elucidation from Infrared Spectra"
_ICLR.cc/2026/Conference — ICLR 2026 Poster_

### Official Review · Reviewer_hhfV · 2025-10-28

**Soundness:** 2
**Presentation:** 2
**Contribution:** 2
**Rating:** 2
**Confidence:** 4

**Summary:**

The paper proposed IR-Agent, a multi-agent AI system for structure elucidation from IR spectra. IR-Agent is composed of three specialized agents: Table Interpreation (TI) expert (extracts local substructures from the IR absorption), Retreiver (Ret) expert (provides global structural cues from retrieved candidates), and Structure Elucidation (SE) expert (integrates both sources of information to infer the final molecular structure). Experiments show improved performance on some experimental IR spectra.

**Strengths:**

1) Application of agents to an IR problem seems novel
2) An ablation study is done on TI and Ret experts, which supports one of the claims.

**Weaknesses:**

1) Lack of iterative correction or self-feedback mechanisms to improve the performance of agents
2) Performance improvements in Table 1, Table 2, and Fig. 2, and 3 are not significant
3) Lack of novelty in agentic system design.
4) No limitations of the method are explained.

Minor issues:
1) Figure 3 appears before Figure 2.
2)  Captions are not descriptive enough to fully grasp the results.
3) What are the numbers in the parentheses in the Figures and Tables?
4) The formal objective of molecular structure elucidation can be explained earlier in the intro.

**Questions:**

1) What is the SOTA non-agentic method to solve the structure elucidation problem in IR?
2) Is there a feedback mechanism in IR-Agents where the Experts can improve themselves in the process?
3) Is there a critique mechanism to monitor the performance of agents?
4) Typically, multiple-agent systems are applied in settings where a design target or goal is imagined and the agents get feedback to take smarter action in the next round. The LLM used in this paper works on solving a one-time task. Is it fair to refer to this system as a multi-agent system? I understand there are two LLMs involved, but I am asking about the "agency". What is making the method an agentic work?
5) Are there innovations in terms of agentic system design?
6) Is there a reason why IR-Agent with GPT-4o and 4o-mini does not outperform the Transformer? Is that expected?

**Details Of Ethics Concerns:**

None.

---

> ### Author Response · Authors · 2025-11-21
>
> **[W1, Q2, Q3]  Regarding iterative-correction & self-feedback & critique mechanism**
> We deeply appreciate the reviewer’s insightful comment regarding the importance of self-feedback and critique mechanisms in agent systems. While we fully agree with this perspective, the current framework of IR-Agent does not yet incorporate these mechanisms. In the following section, we explain the challenges that made applying self-critique/feedback strategies difficult within this particular setting.
> According to [1], LLMs generally struggle to achieve meaningful self-correction without external feedback, and even when feedback is provided, performance improvements can be limited or sometimes detrimental. In the case of IR-Agent, which  elucidate molecular structures solely from IR spectra, the task is inherently more challenging than typical LLM-based reasoning problems, since direct and informative external feedback is required for each subtask. For example, in the Table Interpretation (TI) expert, the LLM itself cannot directly identify the correct peaks or map them to appropriate substructures from raw IR spectra. Instead, it relies on curated tools—such as IR peak table assigners and IR translators—to generate candidate substructure information. However, there are no ground-truth labels for peak–substructure correspondence, since even human chemists infer such mappings based on experience and intuition rather than explicit annotations. Similarly, the Retriever (Ret) expert is required to extract global structural information from candidate SMILES, but again, no labeled data or explicit examples can be provided for supervision.
> As a result, self-feedback or self-correction signals from the LLM itself are difficult to define or exploit effectively.
> We fully agree with the reviewer that designing explicit feedback criteria or structured self-evaluation signals for each expert could be a promising direction to further advance IR-Agent in the future.
>
> [1] Huang, Jie, et al. "Large language models cannot self-correct reasoning yet." ICLR(2024).
>
> **[W2] Regarding significance of performance improvement**
> In accordance with the reviewer’s comment, we conducted t-tests (p-value < 0.05) on all tables and figures to assess the statistical significance of the performance gains, and we report the resulting p-values below. While a few low-K metrics—specifically Table 1 Top-1/3 and Table 2 (carbon count Top-3/5)—do not exhibit statistically significant improvements, the majority of comparisons consistently show significant gains. Notably, Table 1 demonstrates clearer improvements at higher-K metrics, aligning well with how IR-based structure elucidation is carried out in real analytical workflows. In practice, chemists evaluate multiple plausible candidate structures rather than relying solely on a single prediction, making Top-5 and Top-10 metrics more representative and practically relevant. In these settings (Table 1: Top-5/10 and Table 2 scaffold & atom types), IR-Agent achieves statistically significant improvements over the Transformer baseline.
> |                           | Top-1                                | Top-3                                | Top-5                                | Top-10                               |
> |---------------------------|---------------------------------------|---------------------------------------|---------------------------------------|----------------------------------------|
> | table1                    | not significant (0.37654)             | not significant (0.15584)             | significant (0.00183)                 | significant (0.00021)                  |
> | table2 (scaffold)         | significant (0.01745)                 | significant (0.01188)                 | significant (0.00824)                 | significant (0.01758)                  |
> | table2 (carbon count)     | significant (0.00741)                 | not significant (0.07980)             | not significant (0.07385)             | significant (0.01114)                  |
> | table2 (atom types)       | significant (0.00656)                 | significant (0.00591)                 | significant (0.00860)                 | significant (0.00643)                  |
> | Figure2 bar plot          | significant (0.01318)                 | significant (0.00106)                 | significant (0.00440)                 | significant (0.02131)                  |

---

> ### Author Response · Authors · 2025-11-21
>
> To verify that performance gains extend beyond exact-match accuracy, we further computed three Tanimoto similarity metrics—MACCS, RDK, and Morgan—between the predicted and ground-truth structures. Since each test instance produces ten candidate SMILES, we selected the prediction with the highest Tanimoto similarity for evaluation. Across all fingerprints, IR-Agent again shows significant and consistent improvements, indicating that the model generates candidates that are chemically more meaningful.
> |                           | Top-1          | Top-3          | Top-5          | Top-10         | MACCS          | RDK            | Morgan          |
> |---------------------------|----------------|----------------|----------------|----------------|----------------|----------------|------------------|
> | Transformer (Table1)      | 0.098 (0.007)  | 0.169 (0.000)  | 0.176 (0.003)  | 0.176 (0.003)  | 0.698 (0.006)  | 0.514 (0.006)  | 0.490 (0.007)    |
> | IR-Agent (Table1)         | **0.103** (0.005)  | **0.178** (0.007)  | **0.199** (0.004)  | **0.216** (0.001)  | **0.770** (0.004)  | **0.596** (0.002)  | **0.549** (0.003)    |
>
> In summary, while certain low-K metrics exhibit non-significant differences, the higher-K results—together with the molecular structure similarity analyses, which more accurately reflect realistic analytical scenarios where multiple structurally similar candidates matter—offer strong and consistent evidence that IR-Agent achieves substantial improvements over the Transformer baseline.
> **[W3, Q4, Q5] Regarding the design of our agent system**
> We appreciate the reviewer’s comment regarding the limited novelty of our agentic system design, and we would like to clarify the motivation behind our architectural choices.
> IR-Agent intentionally adopts a deterministic, fixed sequence of three expert LLM agents, rather than relying on dynamic decision-making. When interpreting IR spectra, human experts consistently consult both IR absorption tables and spectral databases, leveraging the complementary information from these two sources. Structure elucidation from IR spectra is therefore fundamentally different from general-purpose tasks such as question answering: the problem is domain-specific, highly challenging, and requires faithfully reproducing expert analytical reasoning. Accordingly, our goal was to design a system that explicitly mirrors the workflow used by domain experts.
> To better understand this design choice, we conducted a comparative experiment using a ReAct-style agent framework, where the LLM autonomously selects actions. In this setup, we used GPT-4o as the backbone model and supplied a set of SMILES candidates generated by the IR Spectra Translator. The available tools included the IR Peak Table Assigner, the IR Spectra Retriever, and a finish tool that allows the agent to terminate tool use and produce its final SMILES output. Following the ReAct agent framework, the agent iteratively performed thought, action, and observation steps, with up to five tool calls permitted.
> |                       | Top-1          | Top-3          | Top-5          | Top-10         |
> |-----------------------|----------------|----------------|----------------|----------------|
> | ReAct Framework       | 0.083 (0.004)  | 0.148 (0.005)  | 0.151 (0.003)  | 0.158 (0.005)  |
> | IR-Agent (Table1)     | **0.093** (0.007)  | **0.153** (0.005)  | **0.177** (0.005)  | **0.204** (0.005)  |
>
> Our results show that IR-Agent outperforms the ReAct-based approach. Although the ReAct agent often invoked both the IR Peak Table Assigner and the IR Spectra Retriever, we observed two recurring failure modes: (1) the agent sometimes chose only the IR Peak Table Assigner, restricting itself to an overly narrow evidence source, and (2) the agent frequently repeated the same tool (e.g., retriever → table → table), yielding final predictions biased toward a single information channel. These behaviors suggest that, for IR-based structure elucidation—which demands substantial domain expertise and careful integration of heterogeneous signals—a deterministic tool-selection process, as implemented in IR-Agent, is more suitable. This fixed expert-driven sequence effectively emulates the domain-analysis workflow and underscores the importance of guided, expert-like reasoning for this task.

---

> ### Author Response · Authors · 2025-11-21
>
> **[W4] No limitations of the method are explained**
>
> Thank you for highlighting this important point. As we had already described in Appendix D (Limitations & Future Work), our method is inherently focused on extracting local structural information based on interpretations from the IR absorption table., our method focuses on extracting local structural information based on interpretations derived from the IR absorption table. However, accurate analysis of IR spectra requires not only considering peak positions but also peak shapes and intensities. Since our framework refines and regenerates SMILES using candidates provided by the IR Spectra Translator, its overall performance is inherently affected by the quality of the Translator. At the same time, the framework is designed to flexibly incorporate diverse forms of chemical information without requiring retraining or architectural changes. Nonetheless, when adapting to new spectral datasets, the IR Spectra Translator itself must still be retrained, as is the case for prior approaches.
>
>
> **[Minor W1]**
> Due to layout constraints, Figure 3 was placed before Figure 2 despite their intended order in the text. As the reviewer noted, we have adjusted the placement to improve readability. We appreciate the reviewer’s suggestion.
>
> **[Minor W2, W3]**
> Thanks to the reviewer’s careful observation, we realized that the caption explanations and the descriptions inside the parentheses were not sufficiently clear from a reader’s perspective. The values in parentheses in Tables 1, 2, and 3 indicate the standard deviation of the performance across three independent runs. For Figure 2(a), it shows how the performance changes with the number of translator outputs (C) processed by IR-Agent. Figure 2(b) shows the performance of IR-Agent when it uses a translator pretrained on large-scale simulated IR spectra and subsequently fine-tuned on experimental spectra. This demonstrates the performance improvement achieved by IR-Agent over the transferred translator alone.
>
> **[Minor W4]**
> We have added the objective of molecular structure elucidation in the introduction, stating that it aims to generate the full SMILES representation of an unknown molecule.
>
> **[Q1] Comparison with SOTA Non-Agentic Methods**
> Patch-Based Self-Attention transformer [2] is the SOTA non-agentic method for the structure elucidation from IR spectra. However, relying solely on a standard transformer architecture, this method lacks the ability to mimic expert analysis procedures. Furthermore, its fixed input design limits adaptability, often necessitating architectural redesign to incorporate diverse types of chemical information.
>
> |                                      | Top-1          | Top-3          | Top-5          | Top-10         | MACCS          | RDK            | Morgan          |
> |--------------------------------------|----------------|----------------|----------------|----------------|----------------|----------------|------------------|
> | Patchbased Self-Attention Transformer [2] | **0.129** (0.003)  | 0.160 (0.002)  | 0.170 (0.003)  | 0.181 (0.004)  | 0.721 (0.006)  | 0.547 (0.008)  | 0.518 (0.007)    |
> | IR-Agent                             | 0.103 (0.005)  | **0.178** (0.007)  | **0.199** (0.004)  | **0.216** (0.001)  | **0.770** (0.004)  | **0.596** (0.002)  | **0.549** (0.003)    |
>
> To ensure a fair comparison, we utilized the official GitHub implementation to evaluate the model on the same test dataset used for our IR-Agent. The results show that while the Patch-Based Self-Attention model achieves higher Top-1 accuracy, IR-Agent outperforms it across all other exact-match metrics and every Tanimoto similarity measure. This demonstrates that our approach yields predictions with superior structural relevance. Furthermore, IR-Agent not only surpasses previous baselines in performance but also offers greater extensibility for integrating various forms of chemical knowledge.
>
> [2] Wu, Wenjin, et al. "Transformer-Based Models for Predicting Molecular Structures from Infrared Spectra Using Patch-Based Self-Attention." The Journal of Physical Chemistry A 129.8 (2025): 2077-2085.
>
> **[Q6] Why do GPT-4o and GPT-4o-mini show lower performance?**
> Our experiments reveal a substantial performance gap depending on the backbone model’s reasoning ability. Using gpt-4o-mini resulted in lower performance than the Transformer baseline across all metrics, whereas gpt-4o achieved improvements and surpassed the baseline only in the Top-10 setting. This indicates that the limited reasoning capacity of smaller or non-reasoning models leads to inadequate cross-source integration, producing more incorrect SMILES. Accordingly, we conclude that advanced reasoning ability (o3-mini) is essential for effective structure elucidation within our agentic framework.

---

> > ### Comment · Reviewer_hhfV · 2025-11-22
> > **Appreciate authors response**
> >
> > I really appreciate the author's response. I will increase my score to 4 because of the updates. I cannot be more positive, as I don't see IR Agent as a leap forward compared to the existing patch-based Transformer, and I don't see the significant novelty in the agentic system (e.g., lack of a feedback mechanism).

---

> ### Author Response · Authors · 2025-12-03
>
> We sincerely appreciate the reviewer’s positive feedback and are glad that the rebuttal and additional experiments have strengthened our results. To further emphasize the superiority of our approach over the Patch-Based Attention Transformer, we trained the patch-based model under the chemical-formula-given setting and used this trained model as the IR-Translator to generate candidates, replacing our original Transformer.
>
> | Model                          | Top-1       | Top-3       | Top-5       | Top-10      | MACCS       | RDK         | Morgan      |
> |--------------------------------|-------------|-------------|-------------|-------------|-------------|-------------|-------------|
> | Patch-Based Self-Attention Transformer | 0.160       | 0.244       | 0.279       | 0.331       | 0.841       | 0.679       | 0.592       |
> | IR-Agent  (C=5, 3 runs)                    | 0.190 (0.001) | 0.303 (0.004) | 0.361 (0.004) | 0.420 (0.002) | 0.841 (0.002) | 0.693 (0.004) | 0.638 (0.004) |
>
>  Even in this setting, IR-Agent achieved remarkable improvements, including a 26.9% gain in Top-10 accuracy, demonstrating its strong capability beyond what the patch-based model alone can offer. Moreover, unlike traditional single-model ML approaches such as the patch-based Transformer, which require substantial redesign and retraining whenever new types of chemical information need to be incorporated, our framework provides a flexible, modular architecture. Through expert modules and prompt-based injection of domain knowledge, IR-Agent can seamlessly integrate diverse chemical information without additional model retraining. We believe this modular flexibility is a key advantage that enables IR-Agent to generalize across tasks and settings.

---

### Official Review · Reviewer_MPaz · 2025-10-30

**Soundness:** 2
**Presentation:** 3
**Contribution:** 1
**Rating:** 4
**Confidence:** 5

**Summary:**

The paper introduces IR-Agent, a multi-agent framework for molecular structure elucidation from infrared (IR) spectra. Each agent is designed to emulate a specific aspect of human expert analysis (e.g., identifying functional groups, interpreting peaks, or integrating contextual cues), and the system combines their outputs to predict the full molecular structure, expressed as a SMILES string. The authors claim that this architecture reflects expert-driven reasoning and offers flexibility to incorporate auxiliary chemical information. Experimental results show moderate performance on benchmark IR datasets and limited gains when additional descriptors (e.g., atom types, carbon count, or scaffolds) are provided.

**Strengths:**

The paper tackles an important and challenging problem: automated molecular structure elucidation from spectroscopy data.
The proposed multi-agent architecture is conceptually interesting, providing a modular framework that could, in principle, improve interpretability and extensibility.
The experiments are clearly reported, including dataset details, model parameters, and top-k metrics.
The paper makes a valid attempt to align model design with expert workflows in chemistry.

**Weaknesses:**

Scientific limitations of the problem setup:
IR spectroscopy alone cannot uniquely determine molecular connectivity or size. As a result, the reported top-k accuracies (<20\%) confirm the fundamental ambiguity of the inverse problem. Even with added information (atom types, scaffolds, carbon counts), improvements are marginal (~3\%), limiting the real-world utility of the method.

Shallow “reasoning”:
While the framework uses large language models (e.g., GPT-o-mini) to emulate reasoning, the qualitative examples mostly show pattern annotations such as “the peak at 1700 cm⁻¹ indicates a C=O group.”
For chemists, this does not constitute genuine reasoning, which would involve cross-checking multiple spectral regions, verifying the chemical plausibility of substructures, and ensuring internal consistency (e.g., in anthracene C-H and C=C will be shifted due to the precess of the other rings).
Thus, the claimed interpretability remains superficial and non-scientific, weakening the paper’s argument that the agents emulate expert analysis.

Evaluation metrics are insufficient:
Only top-k accuracy is reported. This metric does not capture the chemical proximity or diversity of predicted molecules. Additional measures such as Tanimoto similarity, token-level accuracy, or functional-group recall would better reflect predictive utility in semi-automated settings.

Given that IR spectra alone cannot determine molecular structure unambiguously, the framework’s current performance is insufficient for realistic applications.
While the topic is relevant and the proposed architecture is creative, the results and qualitative analyses do not convincingly demonstrate meaningful chemical reasoning or practical utility. The low predictive accuracy, combined with the lack of interpretability beyond surface-level annotations, limits the paper’s contribution to the community.

**Questions:**

Did the authors use canonical SMILES when computing top-k accuracy?
What happens when the molecule is not included in the expert knowledge base or training dataset?
Could the authors include Tanimoto similarity, token-level, or functional-group metrics to quantify the chemical diversity of predicted candidates?
Can the authors provide examples of inter-agent reasoning beyond simple peak annotation (e.g., how agents collaborate to ensure structural consistency)?

---

> ### Author Response · Authors · 2025-11-21
>
> **[W1, Q5] Regarding the setting and performance**
> We fully agree with the reviewer that determining a unique molecular structure from IR spectra alone is inherently ambiguous, which results in low performance. As the reviewer correctly noted, IR spectroscopy alone can make it difficult to predict the exact molecular structure.  Nevertheless, we intentionally design our setup without providing the ground-truth chemical formula, even though including it would undoubtedly make the task easier.  Our decision is motivated by three practical considerations: obtaining a reliable formula is often non-trivial in real experimental settings. In particular, while mass spectrometry (MS) is commonly used, it is (1) costly and time-consuming [1], (2) challenging to interpret [2], and (3) frequently insufficient for deriving an exact formula [3, 4]. Under these constraints, training or evaluating a model that requires a ground-truth formula would be impractical, as it would break down whenever the exact formula is unavailable or uncertain. A more detailed justification for this choice of experimental setting is provided in Appendix A.2 (Analysis Setting Details).
>
>
> **Table1**
> |                                      | Top-1          | Top-3          | Top-5          | Top-10         | MACCS          | RDK            | Morgan          |
> |--------------------------------------|----------------|----------------|----------------|----------------|----------------|----------------|------------------|
> | Patchbased Self-Attention Transformer [5] | **0.129** (0.003)  | 0.160 (0.002)  | 0.170 (0.003)  | 0.181 (0.004)  | 0.721 (0.006)  | 0.547 (0.008)  | 0.518 (0.007)    |
> | IR-Agent                             | 0.103 (0.005)  | **0.178** (0.007)  | **0.199** (0.004)  | **0.216** (0.001)  | **0.770** (0.004)  | **0.596** (0.002)  | **0.549** (0.003)    |
>
> The state-of-the-art non-agentic baseline for IR-based structure elucidation—the Patch-Based Self-Attention Transformer [5], which leverages a sophisticated patch-based attention mechanism and two data-augmentation strategies—performs strongly when the ground-truth chemical formula is provided. However, its performance drops sharply under our experimental setting where no formula is given. For a fair comparison, we used the official GitHub implementation and evaluated the model on the same test set as IR-Agent. Our results show that even this SOTA model attains low absolute accuracy, highlighting the fundamental difficulty of the task. Despite being slightly weaker in Top-1 accuracy, IR-Agent surpasses the baseline on all other exact-match metrics as well as every Tanimoto similarity measure. These results indicate that IR-Agent produces predictions that are substantially more structurally meaningful.
>
> Additionally, in response to the reviewer’s point about the limited improvement when additional chemical information is provided, we conducted a detailed analysis of the scaffold-given scenario. Since the scaffold represents the core skeleton of a SMILES structure, supplying it should, in principle, lead to notable performance gains. However, the magnitude of this improvement depends heavily on the evaluation setting, including the exact-match metric and the dataset’s composition. To investigate this, we first examine whether the model effectively leverages scaffold information by measuring the Top-k accuracy of containing the ground-truth scaffold (Table 2). We then explain why this benefit does not fully translate into higher exact-match accuracy.
>
> **Table 2**
> |         | Has scaffold structure                |               |               |               | Exact match of full structure        |               |               |               |
> |---------------------------|----------------------------------------|---------------|---------------|---------------|----------------------------------------|---------------|---------------|---------------|
> |                           | Top-1          | Top-3          | Top-5          | Top-10         | Top-1          | Top-3          | Top-5          | Top-10         |
> | Transformer               | 0.571 (0.004)  | 0.659 (0.004)  | 0.676 (0.006)  | 0.681 (0.009)  | 0.098 (0.007)  | 0.169 (0.000)  | 0.176 (0.003)  | 0.176 (0.003)  |
> | IR-Agent                  | 0.626 (0.003)  | 0.702 (0.002)  | 0.733 (0.001)  | 0.758 (0.001)  | 0.103 (0.005)  | 0.178 (0.007)  | 0.199 (0.004)  | 0.216 (0.001)  |
> | IR-Agent with Scaffold    | 0.931 (0.006)  | 0.956 (0.002)  | 0.961 (0.001)  | 0.967 (0.000)  | 0.118 (0.006)  | 0.208 (0.009)  | 0.232 (0.008)  | 0.258 (0.010)  |

---

> ### Author Response · Authors · 2025-11-21
>
> **How IR-Agent utilizes additional chemical data**
>
> For the Transformer baseline, a large portion of outputs already contain the correct scaffold even without explicit supervision: Top-1 is 57.1% and Top-10 is 68.1%. IR-Agent improves upon this, achieving a Top-1 of 62.6% and a Top-10 of 75.8%, showing that it produces structurally more accurate predictions and that many outputs include the correct scaffold even without explicit guidance. When scaffold information is explicitly provided (IR-Agent w/ Scaffold), we observe consistent gains both in the proportion of predictions containing the correct scaffold and in the exact-match accuracy of the full structure across all Top-k levels.
>
> In the scaffold-given setting, the “Has scaffold structure” metric reaches approximately 96.7% at Top-10, though it does not reach 100%. (A perfect score may be unattainable because IR-Agent integrates information from multiple sources—including the given scaffold, retrieved SMILES candidates, and IR table outputs—so some scaffold details may be modified during this aggregation process.) Overall, these results indicate that the model effectively incorporates scaffold information, contributing to performance improvements in several cases. Notably, we observe gains in both the “Has scaffold structure” and “Exact match of full structure” metrics, suggesting that the model can integrate scaffold information while still producing accurate and valid SMILES outputs.
>
> **Why Providing the Scaffold Does Not Necessarily Lead to Higher Exact-Match Accuracy**
> It is important to note, however, that although the proportion of outputs containing the correct scaffold is very high at Top-10 (96.7%) when the scaffold is provided, the improvement in exact-match accuracy remains relatively modest. While the scaffold provides strong guidance for the overall molecular framework, predicting the exact full structure also requires detailed information about substructures, atom placements, and atom counts. Although the model can generate outputs that incorporate the provided scaffold—leading to observable gains—achieving substantial improvements in exact-match performance is inherently difficult without additional structural detail. Further progress may thus require more specific information, such as precise substructural patterns or atom-level configurations.
>
> Additionally, in our test set, benzene comprises 51% of all molecules, and the total number of unique scaffolds is 144. Because such a large proportion of molecules in both datasets share the benzene scaffold, scaffold information alone often provides limited discriminative power for full-structure prediction. In these cases, additional structural details beyond the scaffold become even more crucial for achieving meaningful improvements in exact-match accuracy.
>
> **[W2, Q2] Regarding the shallow reasoning**
> We agree with the concern that IR-Agent may appear to perform shallow reasoning, such as merely annotating peaks based on the IR table.  As the reviewer noted in the anthracene example, because IR-Agent receives only the mapped information produced by the Peak Assigner, it can incorrectly extract irrelevant peaks and assign incorrect substructures in such cases.
>
> To mitigate this limitation and move beyond simple peak-to-substructure table matching, we designed a prompt for the Table Interpretation (TI) Expert that explicitly instructs the agent to:
>  (1) compare the IR Peak Table Assigner’s output with the SMILES candidate set CCC,
>  (2) identify substructures that are supported by both sources, and
>  (3) generate a confidence level and a concise rationale for each identified substructure (e.g., substructure → confidence → brief rationale).
>
> This mechanism enables IR-Agent to reason about cases where multiple substructures produce absorptions in the overlapped wavenumber region. As a result, the agent improves the robustness and reliability of its table-based interpretation for the target spectra. Additional reasoning examples from TI expert are provided below.

---

> ### Author Response · Authors · 2025-11-21
>
> **[Ex1]**
> **Peaks wavenumber**: [814.0, 1502.0, 2970.0]
> **Table interpretation mapping from the IR absorption table:**
> Peak near (840, 790) cm⁻¹ is associated with trisubstituted alkene (C=C).
> Peak near (1550, 1500) cm⁻¹ is associated with nitro compound (N-O).
> Peak near (3100, 2900) cm⁻¹ is associated with carboxylic acid (O-H).
> Peak near (3200, 2700) cm⁻¹ is associated with alcohol (O-H).
> Peak near (3000, 2800) cm⁻¹ is associated with amine salt (N-H).
> Peak near (3080, 2840) cm⁻¹ is associated with alkane (C-H).
> **Translator output:**
> ['Cc1ccc(C)c(C)c1', 'CCc1ccc(C)c(C)c1', 'CCc1ccc(C)cc1']
> **Answer SMILES**: CCc1ccc(C)cc1C
>
> **Table Agent Reasoning:**
> Alkane (C–H) → High confidence → Each SMILES shows saturated alkyl groups (methyl and/or ethyl substituents) that produce stretching absorptions in the 3080–2840 cm⁻¹ region as noted in the IR interpretation.
> None of the other IR‐assigned substructures (trisubstituted alkene, nitro, carboxylic acid, alcohol, amine salt) appear in these SMILES.
> **Explanation for the Ex1**:
> Although the three detected peaks overlapped with multiple wavenumber regions and initially corresponded to six possible substructures in the IR absorption table, the TI expert validated these assignments against the translator outputs and concluded that only the alkane absorption in the 3080–2840 cm⁻¹ region was supported, while the other substructures were not applicable. This is consistent with the ground-truth SMILES, which indeed contains an alkane.
>
> **[Ex2]**
> **Peaks wavenumber**: [1350.285, 2060.1487]
> **Table interpretation mapping from the IR absorption table:**
> Peak near (1372, 1290) cm⁻¹ is associated with nitro compound (N-O).
>
> Peak near (1420, 1330) cm⁻¹ is associated with alcohol (O-H).
>
> Peak near (1390, 1310) cm⁻¹ is associated with phenol (O-H).
>
> Peak near (1372, 1335) cm⁻¹ is associated with sulfonate (S=O).
>
> Peak near (1370, 1335) cm⁻¹ is associated with sulfonamide (S=O).
>
> Peak near (2140, 1990) cm⁻¹ is associated with isothiocyanate (N=C=S).
> **Translator output:**
> ['CC(C)c1ccc(C#N)cc1', 'CC(C)c1ccc(CN=C=S)cc1', 'C1=CCC1']
> **Answer SMILES**: Cc1ccc(CCN=C=S)cc1
>
> **Table Agent:**
> Isothiocyanate → high confidence → "CC(C)c1ccc(CN=C=S)cc1" exhibits an N=C=S group, matching the IR peak between 2140–1990 cm⁻¹.
> (No IR‐assigned substructures are found in "CC(C)c1ccc(C#N)cc1" or "C1=CCC1" based on the provided mapping.)
> **Explanation for the Ex2**:
> Although the two detected peaks overlapped with multiple wavenumber regions and initially matched six possible substructures in the IR absorption table, the TI expert validated these assignments against the translator outputs and reasoned that only the isothiocyanate group corresponding to the 2140–1990 cm⁻¹ region was present, while the other substructures were not applicable. This is consistent with the ground-truth SMILES, which indeed contains an isothiocyanate group.
>
>
> **[Ex3]**
> **Peaks wavenumber**: [2934.0]
> **Table interpretation mapping from the IR absorption table:**
> Peak near (3100, 2900) cm⁻¹ is associated with carboxylic acid (O-H).
>
> Peak near (3200, 2700) cm⁻¹ is associated with alcohol (O-H) .
>
> Peak near (3000, 2800) cm⁻¹ is associated with amine salt (N-H).
>
> Peak near (3080, 2840) cm⁻¹ is associated with alkane (C-H).
>
> **Translator output:**
> ['CCCCCCCCCCCCCCCCCCc1ccccc1', 'CCCCCCCCCCCCCCCCCCCCCc1ccccc1', 'CCCCCCCCCCCCCCCCCCCc1ccccc1']
> **Answer SMILES**: CCCCCCCCCCCCCCCCCCc1ccccc1
>
> **Table Agent:**
> alkane (C-H) → very high → The SMILES depict long saturated alkyl chains (with a benzene substituent) that reliably account for the alkane C–H stretch absorption near 3080–2840 cm⁻¹; no carboxylic acid, alcohol, or amine group is present.
> **Explanation for Ex3**:
> The single detected peak overlapped with multiple wavenumber regions and initially matched four possible substructures in the IR absorption table. However, after validating these assignments against the translator outputs, the TI expert concluded that only the alkane group corresponding to the 3080–2840 cm⁻¹ region was present, while the other substructures were not applicable. This is consistent with the ground-truth SMILES, which indeed contains an alkane.
>
> Collectively, these examples suggest that while the IR-Agent does not explicitly account for peak shape or intensity—which can lead to misinterpretation—it effectively mitigates this limitation by leveraging the translator’s outputs as a reference for cross-checking and substructure assignment.
>
> **[W3, Q3] Regarding the evaluation metrics**
> To confirm that the improvements are not limited to exact-match accuracy, we additionally measured three Tanimoto similarity scores—MACCS, RDK, and Morgan—between the predicted and ground-truth molecules. Because each test sample yields ten candidate SMILES, we evaluated the one with the highest similarity for each case.

---

> ### Author Response · Authors · 2025-11-21
>
> |                           | Top-1          | Top-3          | Top-5          | Top-10         | MACCS          | RDK            | Morgan          |
> |---------------------------|----------------|----------------|----------------|----------------|----------------|----------------|------------------|
> | Transformer (Table1)      | 0.098 (0.007)  | 0.169 (0.000)  | 0.176 (0.003)  | 0.176 (0.003)  | 0.698 (0.006)  | 0.514 (0.006)  | 0.490 (0.007)    |
> | IR-Agent (Table1)         | **0.103** (0.005)  | **0.178** (0.007)  | **0.199** (0.004)  | **0.216** (0.001)  | **0.770** (0.004)  | **0.596** (0.002)  | **0.549** (0.003)    |
>
> Across all fingerprint types, IR-Agent consistently achieves strong gains, demonstrating that it produces candidates with greater chemical plausibility and structural relevance.
>
> **[W4] Regarding the utility and reasoing**
> Although structure elucidation from only an IR spectrum is inherently challenging and highly ambiguous, we believe that this setting is important and worth pursuing. As discussed earlier, obtaining an exact chemical formula for an unknown target is not always feasible, and sequence-to-sequence models such as Transformers rely on strict input–output mappings. When the chemical formula provided is incorrect or unavailable, such models struggle to operate properly. Moreover, when additional information must be integrated, these single-model approaches typically require a full redesign, retraining, and revalidation of the entire architecture, which is impractical. To address these issues, we adopt an LLM-agent framework in which each subtask is handled by a dedicated module, and their outputs are combined to generate the final prediction.
>
> At the same time, our goal is to enable the LLM agent to reason about structural elucidation in a manner resembling a human chemist. Because an LLM cannot directly interpret an IR spectrum as an image or raw signal, it cannot perform holistic spectrum-level reasoning the way chemists do. This limitation becomes particularly problematic when multiple peaks overlap and several substructures might correspond to the same wavenumber region. To partially mitigate this issue and promote more reliable table-based reasoning, we designed a mechanism in which the agent cross-checks the IR Peak Table Assigner’s mapped substructures with the Translator’s candidate outputs. Through this cross-check reasoning, the agent can extract plausible local substructure information even in ambiguous or overlapping regions.
>
> **[Q1] Regarding the  computing Top-K metrics**
> We compute the Top-K accuracy after converting the SMILES into their InChI representations, which offers a more rigorous and consistent basis for molecular identity comparison than canonical SMILES matching.
>
> **[Q2] Regarding the coverage of knowledge base**
> | IR-Agent   | Top-1          | Top-3          | Top-5          | Top-10         | MACCS          | RDK            | Morgan          |
> |------------|----------------|----------------|----------------|----------------|----------------|----------------|------------------|
> | Transformer | 0.061 (0.004) | 0.120 (0.002)  | 0.133 (0.001)  | 0.133 (0.001)  | 0.690 (0.008)  | 0.491 (0.008)  | 0.465 (0.009)    |
> | IR-Agent    | 0.071 (0.005) | 0.132 (0.004)  | 0.147 (0.002)  | 0.164 (0.004)  | 0.765 (0.006)  | 0.575 (0.005)  | 0.524 (0.008)    |
>
> A knowledge base cannot encompass all possible target molecules, so the ability to elucidate structures for materials that are not present in the database is essential. To assess whether IR-Agent can handle such scenarios, we evaluated its performance on spectra whose SMILES are completely absent from the retrieval DB. As shown in the table, even when the target molecule is not included in the knowledge base (706/906 cases), IR-Agent still demonstrates improved generalization performance on entirely unseen materials.

---

### Official Review · Reviewer_u4dq · 2025-11-05

**Soundness:** 3
**Presentation:** 2
**Contribution:** 2
**Rating:** 4
**Confidence:** 4

**Summary:**

This paper proposes **IR-Agent**, a multi-agent framework for molecular structure elucidation from IR spectra. It consists of three components: the **Table Interpretation Expert**, responsible for retrieving functional groups; the **Retriever Expert**, which retrieves potential molecules; and the **Structure Elucidation Expert**, which integrates all information to infer the corresponding SMILES. Experimental results show that the proposed framework outperforms existing methods, although several key limitations remain.

**Strengths:**

The reasoning strategy of the proposed **IR-Agent** is both reasonable and consistent with that of human experts. Experimental validation is comprehensive, providing evidence of its superiority and the soundness of its architecture.

**Weaknesses:**

The framework was evaluated on a real experimental dataset; however, the overall accuracy of structure inference remains low, even when additional chemical information is provided, **making it far from suitable for practical applications**.

**Questions:**

+ In the **Task Description section**, the IR spectrum is represented as a one-dimensional array. However, different molecules possess distinct functional groups that are reflected at different wavenumbers on the x-axis, **often spanning different ranges**. How can a 1-D array adequately align these patterns and capture the variations in absorbance energy across such differing wavenumber ranges?
+ The **Table Interpretation Expert** is crucial because knowing the functional groups can almost directly determine the molecular structure. The current agent appears to make decisions based on relatively coarse wavenumber intervals, but is this level of accuracy sufficient? A comparison with traditional software such as **Omnic (Thermo Nicolet)** is important, as this is a highly complex problem involving baseline preprocessing, peak location detection, and other subtleties.
+ It is unclear whether the current IR-Agent is capable of handling samples with multiple molecular components. For mixtures, peak overlapping presents a major challenge that should be explicitly discussed. If the method is not designed for such cases, the applicable problem scope should be clearly defined in the paper.
+ In the **Experimental Setup section**, the training IR spectra are used as the database for the Retriever Expert. This raises an important concern about database coverage: does the retrieval database sufficiently include the molecular structures present in the test set? If the database does not fully cover the test molecules, retrieval in step 2 may frequently return incorrect candidates. The authors must therefore clarify whether and how such incorrect retrievals impact the downstream Structure Elucidation Expert (step 3). Conversely, if the database does fully cover the test structures and retrieval accuracy is high, it is important to quantify the incremental contribution of the Structure Elucidation Expert: to what extent does it improve final SMILES inference beyond what a strong Retriever alone (such as Omnic ) would achieve?
+ While the paper reports good performance on real experimental patterns, one critical issue is the low overall accuracy. A survey of results from other published studies on the same or similar tasks would be helpful for context and reference for the reader.

---

> ### Author Response · Authors · 2025-11-21
>
> **[W1, Q5] Regarding the low performance**
> We fully acknowledge the reviewer’s concern regarding the relatively low performance of IR-Agent. Since our experimental setting relies solely on the IR spectrum—without access to any chemical formula—the structure elucidation task becomes considerably more challenging. Although providing the chemical formula would certainly make the task easier, we intentionally adopt the more realistic setting of not assuming that the ground-truth formula is always available. Relying on the chemical formula as a required model input is problematic in practice. Specifically, although mass spectrometry (MS) is commonly used, it is (1) costly and time-consuming [1], (2) difficult to interpret [2], and (3) still leaves the derivation of an exact formula as a highly non-trivial challenge [3, 4]. Under such circumstances, training and evaluating a model that depends on a ground-truth formula would be impractical, as it would fail whenever the true formula is unavailable or inaccurate.
> (The more detailed rationale for adopting this setting is explained  in Appendix A.2 Analysis Setting Details.)
>
> **Table1**
> |                                      | Top-1          | Top-3          | Top-5          | Top-10         | MACCS          | RDK            | Morgan          |
> |--------------------------------------|----------------|----------------|----------------|----------------|----------------|----------------|------------------|
> | Patchbased Self-Attention Transformer [5] | **0.129** (0.003)  | 0.160 (0.002)  | 0.170 (0.003)  | 0.181 (0.004)  | 0.721 (0.006)  | 0.547 (0.008)  | 0.518 (0.007)    |
> | IR-Agent                             | 0.103 (0.005)  | **0.178** (0.007)  | **0.199** (0.004)  | **0.216** (0.001)  | **0.770** (0.004)  | **0.596** (0.002)  | **0.549** (0.003)    |
>
> The state-of-the-art non-agentic baseline for IR-based structure elucidation, the Patch-Based Self-Attention Transformer [5], which incorporates an advanced patch-based attention mechanism along with two forms of data augmentation, performs well in the setting where the ground-truth chemical formula is provided. However, its performance degrades substantially under our experimental setting. For a fair comparison, we used the official GitHub implementation and evaluated the model on the same test dataset as IR-Agent. Our results show that even this non-agentic SOTA model achieves low absolute accuracy, underscoring the inherent difficulty of the task. Nevertheless, although IR-Agent is slightly behind in Top-1 accuracy, it outperforms the baseline across all other exact-match metrics and every Tanimoto similarity measure. These findings demonstrate that our approach yields predictions with substantially higher structural relevance.
>
> Additionally, regarding the reviewer’s concern about the relatively low performance even when additional chemical information is provided, we agree with this point and conducted a detailed analysis of the model under the scaffold-given scenario. Since the scaffold represents the core skeleton of a SMILES structure, providing it should, in principle, yield a substantial performance improvement. However, the extent of this improvement is highly dependent on the evaluation setting, including the exact-match criterion and the dataset composition. To examine this, we first evaluate whether the model is actually leveraging scaffold information by measuring the Top-k accuracy of containing the ground-truth scaffold (Table 2). We then further explain why this advantage does not fully translate into higher exact-match performance.
>
> |         | Has scaffold structure                |               |               |               | Exact match of full structure        |               |               |               |
> |---------------------------|----------------------------------------|---------------|---------------|---------------|----------------------------------------|---------------|---------------|---------------|
> |                           | Top-1          | Top-3          | Top-5          | Top-10         | Top-1          | Top-3          | Top-5          | Top-10         |
> | Transformer               | 0.571 (0.004)  | 0.659 (0.004)  | 0.676 (0.006)  | 0.681 (0.009)  | 0.098 (0.007)  | 0.169 (0.000)  | 0.176 (0.003)  | 0.176 (0.003)  |
> | IR-Agent                  | 0.626 (0.003)  | 0.702 (0.002)  | 0.733 (0.001)  | 0.758 (0.001)  | 0.103 (0.005)  | 0.178 (0.007)  | 0.199 (0.004)  | 0.216 (0.001)  |
> | IR-Agent with Scaffold    | 0.931 (0.006)  | 0.956 (0.002)  | 0.961 (0.001)  | 0.967 (0.000)  | 0.118 (0.006)  | 0.208 (0.009)  | 0.232 (0.008)  | 0.258 (0.010)  |

---

> ### Author Response · Authors · 2025-11-21
>
> **How IR-Agent incorporate additional chemical data**
> For the Transformer baseline, a large proportion of outputs already contain the correct scaffold even without explicit supervision: Top-1 is 57.1% and Top-10 is 68.1%. IR-Agent further improves this, achieving a Top-1 of 62.6% and a Top-10 of 75.8%, demonstrating that it produces structurally more accurate predictions than the Transformer and that a substantial portion of its outputs also contain the correct scaffold without explicit guidance. When scaffold information is explicitly provided (IR-Agent w/ Scaffold), we observe consistent improvements in both the proportion of outputs containing the correct scaffold and the exact match of the full structure across all Top-k levels.
>
> In the scaffold-given setting, the “Has scaffold structure” metric reaches approximately 96.7% at Top-10, though it does not reach 100%. (A perfect score may not be attainable because IR-Agent integrates information from multiple components—including the given scaffold, retrieved SMILES candidates, and IR table interpretations—so some structural details of the original scaffold may be modified during this aggregation process.) Overall, the results indicate that the model effectively incorporates scaffold information into its predictions, contributing to performance gains in several cases. Notably, improvements are observed in both the “Has scaffold structure” and “Exact match of full structure” metrics, indicating that the model can integrate scaffold information while still generating accurate and valid SMILES outputs.
>
> **Why scaffold is not always helpful in exact-match scenarios**
> It is worth noting, however, that although the proportion of outputs containing the correct scaffold is very high at Top-10 (96.7%) when the scaffold is provided, the improvement in exact match performance remains modest. While the scaffold is essential for guiding the overall structure, accurate prediction of the full molecule also depends on additional factors such as detailed substructures, atom placements, and atom counts. Although the model is able to generate SMILES outputs that incorporate the given scaffold—resulting in observable improvements—it is inherently difficult to achieve substantial gains in exact match accuracy without supplementary information describing these finer structural details. Meaningful further improvements will likely require more specific information, such as precise substructural patterns or atom-level arrangements.
>
> Additionally, in our test set, benzene comprises 51% of all molecules, and the total number of unique scaffolds is 144. Given that such a large fraction of molecules in both datasets share this common benzene scaffold, scaffold information alone may not always offer strong discriminative power for full-structure prediction. In these cases, additional structural details beyond the scaffold become even more critical for achieving significant improvements in exact match performance.
>
> [1] Vas, György, and Karoly Vekey. "Solid‐phase microextraction: a powerful sample preparation tool prior to mass spectrometric analysis." Journal of mass spectrometry 39.3 (2004): 233-254.
> [2] Rolland, Amber D., and James S. Prell. "Approaches to heterogeneity in native mass spectrometry." Chemical reviews 122.8 (2021): 7909-7951.
> [3] Böcker, Sebastian, and Kai Dührkop. "Fragmentation trees reloaded." Journal of cheminformatics 8.1 (2016): 5.
> [4] Goldman, Samuel, et al. "MIST-CF: chemical formula inference from tandem mass spectra." Journal of Chemical Information and Modeling 64.7 (2023): 2421-2431.
> [5] Wu, Wenjin, et al. "Transformer-Based Models for Predicting Molecular Structures from Infrared Spectra Using Patch-Based Self-Attention." The Journal of Physical Chemistry A 129.8 (2025): 2077-2085.
>
> **[Q1] Regarding the task description**
> In the Task Description section, the IR spectrum is represented as a one-dimensional array. However, different molecules possess distinct functional groups that are reflected at different wavenumbers on the x-axis, often spanning different ranges. How can a 1-D array adequately align these patterns and capture the variations in absorbance energy across such differing wavenumber ranges?
>
> To clarify how the IR spectrum is represented, and following Na (2024)[2], we first apply polynomial interpolation over a fixed wavenumber range (500–4000 cm⁻¹) to obtain a structured and uniformly sampled spectrum for every molecule, resolving inconsistencies in the number of absorbance points across samples. Therefore, since the $n$-th index of the array consistently corresponds to the same wavenumber across all samples, the positional alignment of functional groups is fully preserved within the 1-D representation. Note that this is widely adopted in ML for infrared spectra [1, 2, 3], enabling ML model can adequately and consistently learn the absorbance patterns.

---

> ### Author Response · Authors · 2025-11-21
>
> [1] Wu, Wenjin, et al. "Transformer-Based Models for Predicting Molecular Structures from Infrared Spectra Using Patch-Based Self-Attention." The Journal of Physical Chemistry A 129.8 (2025): 2077-2085.
> [2] Na, Gyoung S. "Deep Learning for Generating Phase-Conditioned Infrared Spectra." Analytical Chemistry 96.49 (2024): 19659-19669.
> [3] Jung, Guwon, Son Gyo Jung, and Jacqueline M. Cole. "Automatic materials characterization from infrared spectra using convolutional neural networks." Chemical Science 14.13 (2023): 3600-3609.
>
> **[Q2] Regarding the importance of TI expert**
> We fully acknowledge the reviewer’s comment regarding the limited interpretability of the TI expert. As suggested, integrating traditional software such as Omnic would indeed be valuable. However, we currently do not have access to such instruments or corresponding ground-truth datasets, which makes it infeasible to incorporate these tools or conduct direct comparison experiments within the scope of this study.
>
> Instead, we focus on improving the quality of peak interpretation by designing a prompt for the TI Expert that instructs the agent to cross-reference the IR Peak Table Assigner outputs with the translator’s SMILES candidates, identify overlapping substructures, and assign confidence levels accompanied by concise rationales (e.g., substructure → confidence → rationale). This design increases the reliability of table-based interpretation for the target spectrum and guides the agent toward producing more accurate substructure assignments.
>
> We include several success-case analyses to demonstrate the TI expert’s reasoning process. In each example, the TI expert processes the extracted peaks, the corresponding table-based substructures assignments, and the translator’s candidate SMILES, ultimately producing refined substructure predictions with confidence assessments.
>
> **[Ex1]**
> **Peaks wavenumber**: [814.0, 1502.0, 2970.0]
> **Table interpretation mapping from the IR absorption table:**
> Peak near (840, 790) cm⁻¹ is associated with trisubstituted alkene (C=C).
> Peak near (1550, 1500) cm⁻¹ is associated with nitro compound (N-O).
> Peak near (3100, 2900) cm⁻¹ is associated with carboxylic acid (O-H).
> Peak near (3200, 2700) cm⁻¹ is associated with alcohol (O-H).
> Peak near (3000, 2800) cm⁻¹ is associated with amine salt (N-H).
> Peak near (3080, 2840) cm⁻¹ is associated with alkane (C-H).
> **Translator output:**
> ['Cc1ccc(C)c(C)c1', 'CCc1ccc(C)c(C)c1', 'CCc1ccc(C)cc1']
> **Answer SMILES**: CCc1ccc(C)cc1C
>
> **Table Agent Reasoning:**
> Alkane (C–H) → High confidence → Each SMILES shows saturated alkyl groups (methyl and/or ethyl substituents) that produce stretching absorptions in the 3080–2840 cm⁻¹ region as noted in the IR interpretation.
> None of the other IR‐assigned substructures (trisubstituted alkene, nitro, carboxylic acid, alcohol, amine salt) appear in these SMILES.
> **Explanation for the Ex1**:
> Although the three detected peaks overlapped with multiple wavenumber regions and initially corresponded to six possible substructures in the IR absorption table, the TI expert validated these assignments against the translator outputs and concluded that only the alkane absorption in the 3080–2840 cm⁻¹ region was supported, while the other substructures were not applicable. This is consistent with the ground-truth SMILES, which indeed contains an alkane.
>
> **[Ex2]**
> **Peaks wavenumber**: [1350.285, 2060.1487]
> **Table interpretation mapping from the IR absorption table:**
> Peak near (1372, 1290) cm⁻¹ is associated with nitro compound (N-O).
>
> Peak near (1420, 1330) cm⁻¹ is associated with alcohol (O-H).
>
> Peak near (1390, 1310) cm⁻¹ is associated with phenol (O-H).
>
> Peak near (1372, 1335) cm⁻¹ is associated with sulfonate (S=O).
>
> Peak near (1370, 1335) cm⁻¹ is associated with sulfonamide (S=O).
>
> Peak near (2140, 1990) cm⁻¹ is associated with isothiocyanate (N=C=S).
> **Translator output:**
> ['CC(C)c1ccc(C#N)cc1', 'CC(C)c1ccc(CN=C=S)cc1', 'C1=CCC1']
> **Answer SMILES**: Cc1ccc(CCN=C=S)cc1
>
> **Table Agent:**
> Isothiocyanate → high confidence → "CC(C)c1ccc(CN=C=S)cc1" exhibits an N=C=S group, matching the IR peak between 2140–1990 cm⁻¹.
> (No IR‐assigned substructures are found in "CC(C)c1ccc(C#N)cc1" or "C1=CCC1" based on the provided mapping.)
> **Explanation for the Ex2**:
> Although the two detected peaks overlapped with multiple wavenumber regions and initially matched six possible substructures in the IR absorption table, the TI expert validated these assignments against the translator outputs and reasoned that only the isothiocyanate group corresponding to the 2140–1990 cm⁻¹ region was present, while the other substructures were not applicable. This is consistent with the ground-truth SMILES, which indeed contains an isothiocyanate group.

---

> ### Author Response · Authors · 2025-11-21
>
> **[Ex3]**
> **Peaks wavenumber**: [2934.0]
> **Table interpretation mapping from the IR absorption table:**
> Peak near (3100, 2900) cm⁻¹ is associated with carboxylic acid (O-H).
>
> Peak near (3200, 2700) cm⁻¹ is associated with alcohol (O-H) .
>
> Peak near (3000, 2800) cm⁻¹ is associated with amine salt (N-H).
>
> Peak near (3080, 2840) cm⁻¹ is associated with alkane (C-H).
>
> **Translator output:**
> ['CCCCCCCCCCCCCCCCCCc1ccccc1', 'CCCCCCCCCCCCCCCCCCCCCc1ccccc1', 'CCCCCCCCCCCCCCCCCCCc1ccccc1']
> **Answer SMILES**: CCCCCCCCCCCCCCCCCCc1ccccc1
>
> **Table Agent:**
> alkane (C-H) → very high → The SMILES depict long saturated alkyl chains (with a benzene substituent) that reliably account for the alkane C–H stretch absorption near 3080–2840 cm⁻¹; no carboxylic acid, alcohol, or amine group is present.
> **Explanation for Ex3**:
> The single detected peak overlapped with multiple wavenumber regions and initially matched four possible substructures in the IR absorption table. However, after validating these assignments against the translator outputs, the TI expert concluded that only the alkane group corresponding to the 3080–2840 cm⁻¹ region was present, while the other substructures were not applicable. This is consistent with the ground-truth SMILES, which indeed contains an alkane.
>
> From these examples, we can infer that although IR-Agent does not explicitly consider peak shapes or intensity—which can lead to misinterpretation—it mitigates this limitation by indirectly leveraging the translator’s outputs as references when cross-checking and assigning substructures.
>
> **[Q3] Regarding the scope of IR-Agent**
> Unlike previous studies, which have typically limited the scope to single materials in the gas phase, we aimed to design our model in a more realistic and unrestricted setting using the NIST dataset. Consequently, our dataset (Total: 9,052) included both single compounds (8,810) and mixtures (242).
> | Type     | # Test | Top-1          | Top-3          | Top-5          | Top-10         | MACCS          | RDK            | Morgan          |
> |----------|--------|----------------|----------------|----------------|----------------|----------------|----------------|------------------|
> | Single   | 886    | 0.105 (0.006)  | 0.183 (0.008)  | 0.204 (0.005)  | 0.221 (0.001)  | 0.775 (0.003)  | 0.603 (0.003)  | 0.556 (0.003)    |
> | Mixture  | 20     | 0.000          | 0.000          | 0.000          | 0.000          | 0.567 (0.026)  | 0.310 (0.043)  | 0.277 (0.027)    |
>
> However, as the reviewer correctly pointed out, peak overlapping in mixtures presents a significant challenge for accurate prediction. Our empirical results showed that while the IR-Agent performs exceptionally well on single materials, it struggles with mixtures due to signal complexity and data scarcity. Therefore, in this revised manuscript, we have explicitly defined the applicable problem scope of the current IR-Agent as 'single molecular components'. We regard the mixture cases as a challenging future direction to extend the agent's capabilities.
>
> **[Q4] Regarding the retrieval database coverage**
> Among the 906 test samples, only about 200 have an identical SMILES entry present in the retrieval database. The NIST dataset we use contains 9,052 spectra collected across diverse phases (56% gas, 20% liquid, and 24% solid), and therefore identical spectra do not exist even for the same molecule, as IR spectra directly reflect the molecular phase and their shapes can differ considerably depending on phase conditions [1].
>
> | IR-Agent   | Top-1          | Top-3          | Top-5          | Top-10         | MACCS          | RDK            | Morgan          |
> |------------|----------------|----------------|----------------|----------------|----------------|----------------|------------------|
> | Transformer | 0.061 (0.004) | 0.120 (0.002)  | 0.133 (0.001)  | 0.133 (0.001)  | 0.690 (0.008)  | 0.491 (0.008)  | 0.465 (0.009)    |
> | IR-Agent    | 0.071 (0.005) | 0.132 (0.004)  | 0.147 (0.002)  | 0.164 (0.004)  | 0.765 (0.006)  | 0.575 (0.005)  | 0.524 (0.008)    |
>
> As the reviewer pointed out, although no identical IR spectra exist, evaluating the generalization ability of IR-Agent in cases where the retrieval database does not cover the target material is essential. To assess this, we examined whether IR-Agent can still outperform the Transformer baseline on spectra whose SMILES are entirely absent from the retrieval DB. As shown in the table, even when the target molecule is not included in the retrieval database (706 out of 906 cases), IR-Agent continues to demonstrate improved generalization performance on completely unseen materials.
>
> [1] Na, Gyoung S. "Deep Learning for Generating Phase-Conditioned Infrared Spectra." Analytical Chemistry 96.49 (2024): 19659-19669.

---

> > ### Comment · Reviewer_u4dq · 2025-11-23
> > **Further Comment**
> >
> > I appreciate the authors’ detailed response and revisions, which are helpful. However, I still find the experimental setting in `Q5` not entirely fair. I understand that excluding the chemical formula reflects a more realistic usage scenario, but I remain unsatisfied with the current validation setup.
> >
> > A more reasonable ablation would incorporate the chemical formula into IR-Agent and compare its performance with the baselines under the same condition. If the authors can provide evidence that IR-Agent achieves better results when the chemical formula is included, I would be willing to reconsider and potentially elevate my assessment.

---

> ### Author Response · Authors · 2025-12-03
>
> Per the reviewer’s additional request, we trained the Patch-Based Self-Attention model under the “chemical-formula-given” setting in our dataset, strictly following the official GitHub implementation. Because IR-Agent is a flexible framework that can incorporate any IR-Translator, we replaced our original Transformer-based translator with the trained Patch-Based model for a fair comparison under the same chemical-formula condition.
>
>
> | Model                          | Top-1       | Top-3       | Top-5       | Top-10      | MACCS       | RDK         | Morgan      |
> |--------------------------------|-------------|-------------|-------------|-------------|-------------|-------------|-------------|
> | Patch-Based Self-Attention Transformer | 0.160       | 0.244       | 0.279       | 0.331       | 0.841       | 0.679       | 0.592       |
> | IR-Agent (C=5, 3 runs)                      | 0.190 (0.001) | 0.303 (0.004) | 0.361 (0.004) | 0.420 (0.002) | 0.841 (0.002) | 0.693 (0.004) | 0.638 (0.004) |
>
>
> As shown in the table, providing the chemical formula indeed makes the task substantially easier. Notably, under this setting, IR-Agent achieves a remarkable performance improvement over the Patch-Based Self-Attention model, including a 26.9% gain in Top-10 accuracy, highlighting the robustness and effectiveness of IR-Agent in structure elucidation regardless of whether the chemical formula is provided or not. Thus, the reviewer-suggested experiment further demonstrates that IR-Agent can yield consistently strong gains across all settings.
>
> We hope that these empirical results will support a positive assessment, as they directly address the reviewer’s concerns and further validate the strengths of our proposed framework.

---

### Official Review · Reviewer_byX6 · 2025-11-07

**Soundness:** 2
**Presentation:** 2
**Contribution:** 3
**Rating:** 4
**Confidence:** 5

**Summary:**

This paper introduces IR-Agent, a novel LLM-based multi-agent framework for molecular structure elucidation from infrared spectra. The system employs three specialized agents that leverage external tools to emulate expert analytical reasoning. While demonstrating promising performance and extensibility through prompt-based integration of chemical knowledge, the approach exhibits notable limitations.

**Strengths:**

The modeling section of this paper demonstrates a innovative and ingeniously engineered system. It successfully integrates the capabilities of LLMs with domain-specific tools, offering a novel solution pathway for scientific computing problems. Its most significant advantage lies in utilizing multi-agent collaboration to mitigate erroneous judgments that may arise from single-agent systems.

**Weaknesses:**

Data Limitations: With only 9,052 experimental data points, the dataset appears insufficient to adequately demonstrate model generalizability, particularly for large language models requiring substantial training data.

Insufficient Baseline Comparisons: The baseline evaluation lacks comprehensive comparisons with contemporary large models (e.g., Llama3, Claude-3-opus) that have been successfully applied in spectral interpretation tasks. Furthermore, the study fails to compare against state-of-the-art Transformer-based methods that might achieve comparable performance through architectural improvements.

Limited Evaluation Metrics: The sole reliance on accuracy as an evaluation metric is inadequate. The absence of structural similarity measures, such as the widely-recognized Fingerprint Tanimoto Similarity from the RDKit library, prevents comprehensive assessment of prediction quality.

Lack of Multi-agent Mechanism Analysis: The paper provides insufficient analysis explaining why the multi-agent approach outperforms single-agent systems. It fails to demonstrate specific scenarios where multi-agent collaboration prevents errors or enhances reasoning robustness beyond mere accuracy improvements.

Translator Dependency Unexplored: The study doesn't investigate how translation errors propagate through the system, leaving the model's sensitivity to the initial candidate SMILES quality unquantified.

**Questions:**

Data Scale and Generalization Capability
This study utilizes 9,052 experimental spectra for validation. Given the complexity of chemical space and the data requirements of large language models, could the authors provide evidence or discussion on how the framework generalizes to broader molecular diversity? For instance, has testing been conducted on external large-scale datasets or structurally unique compounds? Cross-validation on specific compound categories is recommended to address this limitation.

Comprehensiveness of Baseline Comparisons
Although the paper compares single-agent variants and basic Transformer models, recent studies (e.g., Guo et al.) have employed larger-scale LLMs (such as Llama-3 and Claude-3) for molecular parsing tasks. Could the authors supplement comparisons with such models? Additionally, have advanced Transformer-based methods (e.g., those incorporating enhanced attention mechanisms) been evaluated to ensure performance improvements are not achievable through simple architectural modifications?

Evaluation Metrics Beyond Accuracy
Relying solely on Top-K accuracy may not fully capture the quality of structural predictions. Would the authors consider incorporating structural similarity metrics (e.g., fingerprint similarity calculated via RDKit) to assess whether "approximately correct" predictions retain chemical significance? This would strengthen the argument for the framework's practical utility.

Mechanism Analysis of Multi-Agent Advantages
The superiority of multi-agent systems is primarily attributed to accuracy improvements. Could the authors provide qualitative case studies or error analyses to specify how task division and collaboration avoid typical failure modes of single-agent systems (e.g., handling ambiguous peaks or reconciling conflicting evidence)? Such analysis would clarify the conceptual advantages of the framework.

Sensitivity Analysis of the Initial Translator
The framework's dependency on the IR spectral translator raises concerns about error propagation. Could the authors quantify robustness by testing performance degradation under noisy or low-quality candidate SMILES? For example, how does the system perform when the correct structure is absent from the initial candidate set? Ablation studies on translator quality could delineate the framework's operational boundaries.

---

> ### Author Response · Authors · 2025-11-21
>
> **[W1, Q1] Regarding the generalization**
> We appreciate the reviewer’s detailed and constructive suggestions. Analyzing the model on large-scale experimental IR spectra is important; however, obtaining such datasets is highly challenging due to their limited availability. To the best of our knowledge, the NIST dataset used in our study is the largest and most widely accessible experimental IR spectral resource currently available. Therefore, as the reviewer suggested, we assessed the model’s generalization ability within this dataset by evaluating its performance across (1) different scaffold frequencies and (2) varying heavy-atom counts, thereby reflecting molecular diversity.
> **Table 1 by scaffold frequency**
> | Model       | Scaffold frequency | Number of data | Top-1 | Top-3 | Top-5 | Top-10 | MACCS | RDK  | Morgan |
> |-------------|---------------|----------------|-------|-------|-------|--------|-------|------|--------|
> | Transformer | >100          | 562            | 0.132 | 0.230 | 0.237 | 0.237  | 0.760 | 0.582 | 0.559  |
> | Transformer | 30<= <100     | 150            | 0.033 | 0.086 | 0.093 | 0.093  | 0.651 | 0.428 | 0.400  |
> | Transformer | <30           | 194            | 0.050 | 0.055 | 0.064 | 0.064  | 0.563 | 0.382 | 0.355  |
> | IR-Agent    | >100          | 562            | 0.132 | 0.233 | 0.255 | 0.273  | 0.813 | 0.652 | 0.603  |
> | IR-Agent    | 30<= <100     | 150            | 0.055 | 0.107 | 0.129 | 0.149  | 0.740 | 0.535 | 0.495  |
> | IR-Agent    | <30           | 194            | 0.055 | 0.075 | 0.093 | 0.101  | 0.670 | 0.481 | 0.435  |
>
>
> Our NIST dataset contains 216 unique scaffolds and exhibits a highly diverse, long-tail distribution: 7 scaffolds appear more than 100 times, 31 appear between 30 and 100 times, and 178 are rare scaffolds that occur fewer than 30 times. Based on these three scaffold-frequency groups, we re-evaluated model performance. For IR-Agent, performance decreased consistently across all metrics as scaffolds became rarer; however, the drop in Tanimoto similarity metrics was smaller than the drop observed in the Top-k exact-match accuracy. This suggests that the agent continues to produce structurally plausible and chemically relevant candidates, even in the long-tail regime where exact matching becomes inherently difficult. Moreover, IR-Agent outperformed the Transformer baseline in every scaffold-frequency group, indicating improved generalization even for rare scaffolds.
>
> **Table 2 by heavy atom counts**
> | Model       | Number of heavy atoms | Number of data | Top-1 | Top-3 | Top-5 | Top-10 | MACCS | RDK  | Morgan |
> |-------------|------------------------|----------------|-------|-------|-------|--------|-------|------|--------|
> | Transformer | <10                    | 166            | 0.179 | 0.287 | 0.301 | 0.301  | 0.713 | 0.542 | 0.549  |
> | Transformer | 10<= <20               | 646            | 0.089 | 0.160 | 0.165 | 0.165  | 0.703 | 0.517 | 0.485  |
> | Transformer | 20<= <50               | 94             | 0.021 | 0.021 | 0.028 | 0.028  | 0.640 | 0.438 | 0.420  |
> | IR-Agent    | <10                    | 166            | 0.189 | 0.303 | 0.327 | 0.353  | 0.780 | 0.620 | 0.604  |
> | IR-Agent    | 10<= <20               | 646            | 0.093 | 0.168 | 0.188 | 0.204  | 0.772 | 0.599 | 0.545  |
> | IR-Agent    | 20<= <50               | 94             | 0.021 | 0.032 | 0.050 | 0.053  | 0.739 | 0.534 | 0.487  |
>
> Additionally, our dataset contains molecules with a mean of 13.4 and a median of 12 heavy atoms. We further re-evaluated performance by grouping molecules into three ranges: fewer than 10 heavy atoms, between 10 and 20, and more than 20. As molecular size increased, performance declined, particularly in the Top-k exact-match metrics. This is expected because exact matching becomes increasingly difficult as the number of tokens to be predicted grows with molecular size. In contrast, the Tanimoto similarity metrics exhibited only modest decreases, demonstrating that IR-Agent remains robust in generating structurally relevant predictions. Across all heavy-atom ranges, IR-Agent again outperformed the Transformer, confirming its superior generalization capability.
>
> **[W2, Q2] Regarding the baseline comparison**
> Thank you for your helpful feedback, which has strengthened our work. The recent study mentioned by the reviewer (Guo et al.) corresponds to [1]; it is a pioneering work that frames molecular structure elucidation as a crossword puzzle, integrates multiple spectroscopic modalities, and provides both a dataset and extensive LLM experiments. Following the reviewer’s suggestion, we conducted additional experiments using Llama-3-8B—as used in [1] Guo et al.—to directly perform our target task of predicting SMILES from IR spectra.

---

> ### Author Response · Authors · 2025-11-21
>
> Since IR spectra consist of thousands of floating-point intensity values, we first downsampled each spectrum to 400 points and normalized intensities to the range 0–99 to make tokenization feasible. We then evaluated both (1) zero-shot performance using the pretrained Llama model and (2) QLoRA-based fine-tuning.
>
> In both settings, the model failed to generate valid SMILES strings and often produced severely corrupted or nonsensical outputs. For the zero-shot case, this outcome is expected because the model has never been exposed to any form of mapping between IR values and molecular structures. For the fine-tuning case, the difficulty appears to stem from the relatively modest dataset size and the challenge of learning meaningful spectral patterns directly from numeric sequences. The downsampling step may also have introduced information loss, further degrading model performance. The dataset used for the Llama experiments is provided below.
>
> `[{'role': 'system', 'content': 'You are an expert for IR spectroscopy, especially for molecular structure elucidation'}, {'role': 'user', 'content': 'Given IR absorbance: IR 0 2 3 5 6 8 9 11 12 14 15 17 18 20 21 23 24 26 27 29 30 32 33... 46 46, generate the corresponding SMILES.'}]`
>
> | Model                                   | Top-1          | Top-3          | Top-5          | Top-10         | MACCS          | RDK            | Morgan          |
> |-----------------------------------------|----------------|----------------|----------------|----------------|----------------|----------------|------------------|
> | Llama-3-8                             | -              | -              | -              | -              | -              | -              | -                |
> | Patchbased Self-Attention Transformer [2] | **0.129** (0.003)  | 0.160 (0.002)  | 0.170 (0.003)  | 0.181 (0.004)  | 0.721 (0.006)  | 0.547 (0.008)  | 0.518 (0.007)    |
> | IR-Agent                                 | 0.103 (0.005)  | **0.178** (0.007)  | **0.199** (0.004)  | **0.216** (0.001)  | **0.770** (0.004)  | **0.596** (0.002)  | **0.549** (0.003)    |
>
> In addition to Llama experiments, we also compared IR-Agent with another Transformer architecture featuring an advanced attention mechanism, as suggested by the reviewer. The Patch-Based Self-Attention Transformer [2] represents the state-of-the-art non-agentic method for IR-based structure elucidation, incorporating a sophisticated patch-level attention design along with two data-augmentation strategies. To ensure a fair comparison, we used the official GitHub implementation and evaluated the model on the same test dataset as IR-Agent. Our results show that although the Patch-Based Self-Attention model attains higher Top-1 accuracy, IR-Agent outperforms it across all other exact-match metrics as well as every Tanimoto similarity measure. This indicates that our approach produces structurally more meaningful and chemically relevant predictions.
>
> [1] Guo, Kehan, et al. "Can llms solve molecule puzzles? a multimodal benchmark for molecular structure elucidation." Advances in Neural Information Processing Systems 37 (2024): 134721-134746.
> [2] Wu, Wenjin, et al. "Transformer-Based Models for Predicting Molecular Structures from Infrared Spectra Using Patch-Based Self-Attention." The Journal of Physical Chemistry A 129.8 (2025): 2077-2085.
>
> **[W3, Q3] Regarding the evaluation metrics**
> |                           | Top-1          | Top-3          | Top-5          | Top-10         | MACCS          | RDK            | Morgan          |
> |---------------------------|----------------|----------------|----------------|----------------|----------------|----------------|------------------|
> | Transformer (Table1)      | 0.098 (0.007)  | 0.169 (0.000)  | 0.176 (0.003)  | 0.176 (0.003)  | 0.698 (0.006)  | 0.514 (0.006)  | 0.490 (0.007)    |
> | IR-Agent (Table1)         | **0.103** (0.005)  | **0.178** (0.007)  | **0.199** (0.004)  | **0.216** (0.001)  | **0.770** (0.004)  | **0.596** (0.002)  | **0.549** (0.003)    |
>
> To assess whether the improvements go beyond exact-match accuracy, we additionally evaluated three Tanimoto similarity metrics—MACCS, RDK, and Morgan—between the predicted and ground-truth molecules. For each test case, we selected the candidate SMILES with the highest similarity among the ten generated predictions. Across all fingerprint based metrics, IR-Agent demonstrates clear and consistent gains, showing that it produces structurally more meaningful and chemically relevant candidates.

---

> ### Author Response · Authors · 2025-11-21
>
> **[W4, Q4] Regarding the advantages of multi-agent systems**
> Thank you for the helpful suggestion regarding providing concrete case studies that illustrate how the single-agent version of IR-Agent fails to perform the task. Below, we present two case studies in which the multi-agent framework successfully produced the correct prediction, while the single-agent version did not.
>
> In the first example, the multi-agent framework correctly assigned the alkane (C–H) functional group and further reasoned—through cross-checking between the table interpretation and the Translator’s outputs—that the trisubstituted alkene, nitro, carboxylic acid, alcohol, and amine salt groups were not consistent with the target structure. In contrast, the single-agent model only rejected the trisubstituted alkene, nitro, and carboxylic acid groups, failing to include alcohol and amine salt in its reasoning. More importantly, it entirely omitted the alkane (C–H) substructure—which should have been identified from the IR table and is indeed present in the ground-truth SMILES—resulting in an incomplete and inconsistent reasoning process. As a result, the multi-agent framework successfully predicted the correct SMILES (“CCc1ccc(C)cc1C”), whereas the single-agent model failed.
>
> In the second example, the multi-agent framework accurately extracted both the Alkyl aryl C–O and methylene (alkane C–H) substructures. However, the single-agent model focused primarily on the retrieved candidates and identified only the Alkyl aryl C–O functionality, neglecting the methylene group. Consequently, it failed to predict the correct SMILES (“OCCOc1ccc2ccccc2c1”).
>
> Across these two case studies, we observe that the multi-agent framework is able to extract fine-grained local substructures by cross-checking evidence between the Translator’s outputs and the IR table interpretation. In contrast, the single-agent model attempts to process the retrieved candidates, table information, and Translator outputs simultaneously, often missing substructures that require more careful table-based reasoning or over-focusing on features in the retrieved candidates. These examples demonstrate that dividing the task among the TI, Ret, and SE experts and aggregating their outputs leads to more reliable and robust reasoning.
>
> **[W5, Q5] Sensitivity analysis of translator**
> Thank you for the insightful feedback. Since IR-Agent relies on the Translator’s SMILES candidate set C to generate its output, its performance is naturally influenced by the quality of these candidates. To analyze this dependency, we conducted two types of experiments.
> (1) When the correct SMILES is present in the Translator’s output, we remove it and evaluate IR-Agent using only the remaining (incorrect) candidates.
>  (2) For each candidate, we randomly delete one token to intentionally degrade its quality and examine how IR-Agent performs with low-quality inputs.
>
> Experiment (1) allows us to assess the agent’s robustness when the correct structure is not included among the candidates, while experiment (2) measures how sensitive the agent is to degraded or noisy SMILES.
>
>
> | Model                        | Top-1          | Top-3          | Top-5          | Top-10         | MACCS          | RDK            | Morgan          |
> |------------------------------|----------------|----------------|----------------|----------------|----------------|----------------|------------------|
> | type1: removing gt SMILES if exists  | 0.029 (0.003)  | 0.056 (0.009)  | 0.087 (0.011)  | 0.113 (0.011)  | 0.764 (0.001)  | 0.574 (0.002)  | 0.513 (0.002)    |
> | type2: randomly deleting      | 0.005 (0.004)  | 0.105 (0.000)  | 0.130 (0.003)  | 0.154 (0.003)  | 0.753 (0.005)  | 0.557 (0.001)  | 0.504 (0.002)    |
> | IR-Agent                     | 0.103 (0.005)  | 0.178 (0.007)  | 0.199 (0.004)  | 0.216 (0.001)  | 0.770 (0.004)  | 0.596 (0.002)  | 0.549 (0.003)    |
>
> As shown in the table, removing the ground-truth SMILES from the candidate list leads to the largest drop in exact-match accuracy, and randomly corrupting the SMILES strings also reduces performance. This is expected because exact-match metrics require token-level correctness, making them highly sensitive to even small perturbations. Nevertheless, the decline in Tanimoto similarity metrics is considerably smaller. Although the exact-match accuracy decreases substantially, IR-Agent still produces structurally relevant predictions, demonstrating robustness in terms of chemical similarity even when candidate quality deteriorates.

---

### Author Response · Authors · 2025-12-03
**Summary of our rebuttal for AC and SAC**

Dear AC and SAC,

Thank you for managing the review process under these unusual circumstances. To help streamline your workload, we offer a concise summary of the reviewers’ feedback and our corresponding rebuttal.  We first summarize our experimental results and responses addressing the common concerns raised by multiple reviewers, followed by a concise summary of our replies to each individual review.

- **Regarding metrics, performance, and the baseline**

First, during the rebuttal period, we incorporated additional evaluations such as Tanimoto similarity metrics and comparisons with the baseline Patch-Based Self-Attention Transformer. We also demonstrated that IR-Agent performs strongly even under the chemical-formula-given setting. Through these additional experiments, we show that IR-Agent outperforms the Patch-Based Self-Attention model, a SOTA approach, under the more challenging setting where the chemical formula is not provided, achieving superior performance across Top-k metrics (Top-3, Top-5, Top-10) as well as all Tanimoto similarity measures.

Furthermore, because IR-Agent is a flexible framework that can utilize any IR-spectrum translator, it can also operate effectively when the ground-truth chemical formula is provided. When using the Patch-Based Self-Attention model as the IR-Translator (C=5 candidates) along with chemical-formula information, IR-Agent achieves a **26.9% improvement in Top-10 accuracy**, demonstrating its strong capability to enhance even advanced baseline models (**Please refer to the table included in our final response to reviewer u4dq**).

---
### **Reviewer byX6** ###
**Regarding the generalization**, we evaluate IR-Agent under two scenarios: (1) different scaffold frequencies and (2) varying heavy-atom counts. From these results, we observe that IR-Agent can propose structurally plausible and chemically relevant candidates even in rare or low-frequency cases.

**Regarding the advantages of multi-agent systems**, we provide concrete case studies that highlight situations where the single-agent version of IR-Agent fails to complete the task. In contrast, the multi-agent framework successfully produces the correct predictions in these scenarios, demonstrating the necessity and effectiveness of the multi-agent design.

**Regarding the sensitivity analysis of the translator**, we designed two experiments. Experiment (1) assesses the agent’s robustness when the correct structure is not included among the candidate SMILES, while experiment (2) evaluates how sensitive the agent is to degraded or noisy SMILES inputs. Together, these analyses allow us to examine how the choice and quality of the translator affect IR-Agent’s performance.

---

### **Reviewer u4dq** ###

During the rebuttal, the reviewer requested additional experiments under the chemical-formula-given setting and stated that “If the authors can provide evidence that IR-Agent achieves better results when the chemical formula is included, I would be willing to reconsider and potentially elevate my assessment.” As demonstrated in our rebuttal response, IR-Agent indeed shows a remarkable performance improvement over the Patch-Based Self-Attention model, including a 26.9% gain in Top-10 accuracy in the chemical-formula-given setting. This result highlights the robustness and effectiveness of IR-Agent in structure elucidation from IR spectra.

**Regarding the importance of the TI expert**, we explain how the TI expert performs reliable interpretation of IR spectra and provide three case studies that support its necessity and contribution.

**Regarding the scope of IR-Agent**, we clarified the applicability of the framework by presenting results on both single-compound and mixture scenarios.

**Regarding the retrieval database coverage**, we experimentally showed that IR-Agent remains effective even when the identical molecule is absent from the retrieval database, demonstrating its ability to generalize beyond explicit database matches.

---

### **Reviewer MPaz** ###

**Regarding the shallow reasoning issue**, we provide three case studies to clarify how IR-Agent mitigates such failures. Since IR-Agent receives only the mapped information produced by the Peak Assigner, it may occasionally extract irrelevant peaks and assign incorrect substructures. The case studies illustrate how the multi-expert design helps address these weaknesses and reduces errors arising from shallow or misled interpretations.

**Regarding the IR only setting**, we further clarify why, despite the inherent difficulty and ambiguity of using only IR spectra, we believe that this setting remains important and valuable to pursue.

**Regarding the retrieval database coverage**, we experimentally demonstrate that IR-Agent performs well even when the retrieval database does not contain an identical molecule to the target IR spectrum, indicating that the system can generalize beyond explicit database matches.

---

> ### Author Response · Authors · 2025-12-03
> **Summary of our rebuttal for AC and SAC**
>
> ### **Reviewer hhfV** ###
>
> During the rebuttal, the reviewer responded positively to our explanations and additional experiments, addressing all concerns except the discussion on the novelty of the agentic design.
>
> **Regarding the lack of iterative-correction, self-feedback, and critique mechanisms**, the reviewer argued that the absence of such mechanisms limits the novelty of our agentic design. In response, we provided a justification for why these mechanisms were not incorporated and clarified the intended design philosophy of IR-Agent.
>
> **Regarding the novelty of the agentic design**, we further explained the motivation behind our architectural choices and supplemented the discussion with an additional comparison to the ReAct framework to clarify the soundness and appropriateness of our design.
>
> **Regarding the concern that no limitations were provided**, we pointed out that the limitations of our method are already detailed in Appendix D and reiterated them for clarity.
>
> **Regarding the remaining minor issues raised by the reviewers**, we revised and clarified the corresponding parts of the manuscript accordingly.
>
> Unfortunately, because of the incident that occurred during the rebuttal phase, the reviewers did not have adequate time to fully assess our responses before the discussion period closed. Nevertheless, the additional experiments with stronger baselines, the expanded evaluation metrics, and the remarkable performance improvements across new experimental settings collectively provide clear evidence of the effectiveness of IR-Agent. In addition, the case studies further demonstrate how the system is capable of performing plausible chemical reasoning in a variety of scenarios. We respectfully hope that our comprehensive responses and supplementary analyses will be considered in your final recommendation. Thank you for your time and thoughtful evaluation.

---

### Meta-Review · Area_Chair_5eVr · 2026-01-09

**Summary:**

The reviewers broadly agree that the problem of molecular structure elucidation from IR spectra is interesting, and that the proposed multi-agent framework is reasonable. The main concerns are:
(1) Low absolute performance and the inherent ambiguity of IR-only structure elucidation;
(2) Insufficient evaluation in the initial submission;
(3) Depth and novelty of the agentic design;
(4) Unclear scope and robustness, including sensitivity to the translator, retrieval database coverage, and handling of mixtures.

**Reviewer Concerns:**

Concerns addressed:
- Multi-agent justification with concrete qualitative case studies, showing failure modes of the single-agent variant and how multi-agent decomposition improves reasoning robustness.
- Evaluation metrics of structural similarity measures were addressed through the addition of multiple Tanimoto similarity metrics (MACCS, RDK, Morgan).
- Baseline comparisons were added against a state-of-the-art Patch-Based Self-Attention Transformer and conducted additional experiments with Llama-3-8B.
- Generalization and robustness were addressed in new analyses by scaffold frequency, heavy-atom count, retrieval-database exclusion, and translator-noise ablations.

Concerns remain:
- Depth and novelty of agentic reasoning; more on engineering side.
- Low absolute accuracy and real-world utility.
- Scientific interpretability in the reasoning traces appear to be at surface-level.

**Reviewer Scores:**

The lowest score (2) was given by Reviewer hhfV, who explicitly said they will increase the score to 4 after rebuttal.

Reviewer MPaz: Unlikely to change score. Although metrics and case studies were added, their core critique about shallow reasoning and limited scientific utility remains

Reviewer byX6 and Reviewer u4dq could be likely to increase their rating.

---

### Decision · Program_Chairs · 2026-01-26

Accept (Poster)